# A defect in myoblast fusion underlies Carey-Fineman-Ziter syndrome

Silvio Alessandro Di Gioia[1,2,3], Samantha Connors[4,*], Norisada Matsunami[5,*], Jessica Cannavino[6,*], Matthew F. Rose[1,2,7,8,9,10,*], Nicole M. Gilette[1,2], Pietro Artoni[1,2,3], Nara Lygia de Macena Sobreira[11], Wai-Man Chan[1,2,3,12], Bryn D. Webb[13], Caroline D. Robson[14,15], Long Cheng[1,2,3], Carol Van Ryzin[16], Andres Ramirez-Martinez[6], Payam Mohassel[17,18], Mark Leppert[5], Mary Beth Scholand[19], Christopher Grunseich[18], Carlos R. Ferreira[16], Tyler Hartman[20], Ian M. Hayes[21], Tim Morgan[4], David M. Markie[22], Michela Fagiolini[1,2,3], Amy Swift[16], Peter S. Chines[16], Carlos E. Speck-Martins[23], Francis S. Collins[16,24], Ethylin Wang Jabs[11,13], Carsten G. Bönnemann[17,18], Eric N. Olson[6], Moebius Syndrome Research Consortium[†], John C. Carey[25], Stephen P. Robertson[4], Irini Manoli[16], Elizabeth C. Engle[1,2,3,8,10,12,26,27]

Multinucleate cellular syncytial formation is a hallmark of skeletal muscle differentiation. Myomaker, encoded by *Mymk (Tmem8c)*, is a well-conserved plasma membrane protein required for myoblast fusion to form multinucleated myotubes in mouse, chick, and zebrafish. Here, we report that autosomal recessive mutations in *MYMK* (OMIM 615345) cause Carey-Fineman-Ziter syndrome in humans (CFZS; OMIM 254940) by reducing but not eliminating MYMK function. We characterize *MYMK*-CFZS as a congenital myopathy with marked facial weakness and additional clinical and pathologic features that distinguish it from other congenital neuromuscular syndromes. We show that a heterologous cell fusion assay *in vitro* and allelic complementation experiments in *mymk* knockdown and *mymk*[insT/insT] zebrafish *in vivo* can differentiate between *MYMK* wild type, hypomorphic and null alleles. Collectively, these data establish that MYMK activity is necessary for normal muscle development and maintenance in humans, and expand the spectrum of congenital myopathies to include cell-cell fusion deficits.

[1] Department of Neurology, Boston Children's Hospital, Boston, Massachusetts 02115, USA. [2] F.M. Kirby Neurobiology Center, Boston Children's Hospital, Boston, Massachusetts 02115, USA. [3] Department of Neurology, Harvard Medical School, Boston, Massachusetts 02115, USA. [4] Department of Women's and Children's Health, Dunedin School of Medicine, University of Otago, Dunedin 9054, New Zealand. [5] Department of Human Genetics, University of Utah School of Medicine, Salt Lake City, Utah 84132, USA. [6] Department of Molecular Biology and Neuroscience, and Hamon Center for Regenerative Science and Medicine, The University of Texas Southwestern Medical Center, Dallas, Texas 75390 USA. [7] Department of Pathology, Boston Children's Hospital, Boston, Massachusetts 02115, USA. [8] Medical Genetics Training Program, Harvard Medical School, Boston, Massachusetts 02115, USA. [9] Department of Pathology, Brigham and Women's Hospital and Harvard Medical School, Boston, Massachusetts 02115, USA. [10] Broad Institute of M.I.T. and Harvard, Cambridge, Massachusetts 02142, USA. [11] McKusick-Nathans Institute of Genetic Medicine, Department of Pediatrics, Johns Hopkins University School of Medicine, Baltimore, Maryland 21205, USA. [12] Howard Hughes Medical Institute, Chevy Chase, Maryland 20815, USA. [13] Department of Genetics and Genomic Sciences, Icahn School of Medicine at Mount Sinai, New York 10029, USA. [14] Department of Radiology, Boston Children's Hospital, Boston, Massachusetts 02115, USA. [15] Department of Radiology, Harvard Medical School, Boston, Massachusetts 02115, USA. [16] Medical Genomics and Metabolic Genetics Branch, National Human Genome Research Institute, National Institutes of Health, Bethesda, Maryland 20892-1477, USA. [17] Neuromuscular and Neurogenetic Disorders of Childhood Section, National Institute of Neurological Disorders and Stroke, National Institutes of Health, Bethesda, Maryland 20892-1477, USA. [18] Neurogenetics Branch, National Institute of Neurological Disorders and Stroke, National Institutes of Health, Bethesda, Maryland 20892-1477, USA. [19] Department of Internal Medicine, University of Utah School of Medicine, Salt Lake City, Utah 84132, USA. [20] Department of Pediatrics, Dartmouth-Hitchcock Medical Center, Geisel School of Medicine, Hanover, New Hampshire 03755-1404, USA. [21] Genetic Health Services New Zealand, Auckland City Hospital, Auckland 1142, New Zealand. [22] Department of Pathology, Dunedin School of Medicine, University of Otago, Dunedin 9054, New Zealand. [23] SARAH Network of Rehabilitation Hospitals, Brasilia 70335-901, Brazil. [24] Office of the Director, National Institutes of Health, Bethesda, Maryland 20892-1477, USA. [25] Department of Pediatrics, University of Utah School of Medicine, Salt Lake City, Utah 84132, USA. [26] Department Ophthalmology, Boston Children's Hospital, Boston, Massachusetts 02115, USA. [27] Department of Ophthalmology, Harvard Medical School, Boston, Massachusetts 02115, USA. * These authors contributed equally to this work. Correspondence and requests for materials should be addressed to J.C.C. (email: John.Carey@hsc.utah.edu) or to I.M. (email: manolii@mail.nih.gov) or to E.C.E. (email: elizabeth.engle@childrens.harvard.edu).

[†] A full list of consortium members appear at the end of the paper.

arey-Fineman-Ziter (CFZS; OMIM 254940) is an eponymous syndrome described in two siblings who had marked bilateral facial weakness, Robin sequence (mandibular hypoplasia, hypoglossia, cleft palate), inability to fully abduct both eyes, characteristic facial dysmorphisms, generalized muscle hypoplasia with hypotonia and relatively mild proximal weakness, failure to thrive, delayed motor milestones, scoliosis, and normal intelligence[1,2]. Reports of children subsequently diagnosed with CFZS have differed from the original siblings by also having intellectual disabilities, seizures, and/or brain malformations or calcifications, raising concerns that they may not represent the same disease entity[2–7]. Many of these subsequent cases had facial weakness and complete absence of ocular abduction, thus meeting diagnostic criteria for Moebius syndrome[8]. In an effort to identify the genetic etiology of CFZS, we enrolled the original CFZS pedigree and additional sibling pairs and simplex cases with similar phenotypes together with their family members into ongoing genetic studies at six medical institutions. By exome and Sanger sequencing, we now identify compound heterozygous or homozygous MYMK (TMEM8C) missense mutations in affected members of four CFZS sibling pairs and one simplex case.

During embryogenesis, myoblast precursors migrate from pre-somitic mesoderm to the site of muscle formation, where they then fuse to form multinucleate myotubes and ultimately myofibers[9,10]. Multinucleated myofibers then confer distinct biomechanical advantages to mature muscles by transducing force between remote skeletal attachments. Myomaker, encoded by Mymk, mediates the fusion of mononuclear myoblasts to form multinucleate myocyte syncytia during muscle development in mouse[11], chick[12] and zebrafish[13,14]. $Mymk^{-/-}$ mice have a paucity of skeletal muscle and die perinatally, while heterozygotes ($Mymk^{+/-}$) are indistinguishable from wild-type (WT) littermates[11]. Moreover, $Mymk^{-/LoxP}$ conditional knockout mice under the control of a muscle satellite cell specific promoter have defective muscle regeneration[15].

Phenotypic evaluation of the affected participants in this study defines MYMK-CFZS as a congenital myopathy with marked facial weakness and additional clinical and pathologic features that distinguish it from other congenital neuromuscular syndromes and from Moebius syndrome. Heterologous cell fusion assays and zebrafish modelling demonstrate that MYMK mutations cause CFZS by reducing but not eliminating MYMK function. Collectively, these data establish that MYMK activity is necessary for normal muscle development and maintenance in humans, and expand the spectrum of congenital myopathies to include cell–cell fusion deficits.

## Results

**Mutations in MYMK cause CFZS.** To identify the genetic etiology of CFZS, three laboratories each independently generated and analysed exome sequence data from one of three unrelated non-consanguineous families, each with two affected siblings with similar phenotypes (Families 1–3, Figs 1a and 2a, Table 1). Exome analysis was performed to identify recessive variants with minor allele frequencies (MAF) of <0.01. Among the genes harbouring recessive variants identified in each family, only MYMK was common to all three (Supplementary Table 1a–c). The affected siblings in USA Family 1 (individuals 1 and 2), the original CFZS pedigree[1,2], inherited MYMK c.271C>A (p.Pro91Thr, rs776566597, M1) and c.553T>C (p.Cys185Arg, M2) from their father and mother, respectively. The affected siblings in New Zealand Family 2 (individuals 3 and 4) and the affected siblings in USA Family 3 (individuals 5 and 6) each inherited M1 and c.298G>A (p.Gly100Ser, M3) from their father and mother,

respectively (Fig. 1a,b and Table 1). Variants and their co-segregation were validated by Sanger sequencing (Supplementary Fig. 1a).

To define the phenotypic heterogeneity among individuals harbouring MYMK variants, MYMK exons and flanking intron-exon boundaries were sequenced in >300 additional probands with congenital facial weakness who were referred with the following diagnoses: CFZS (1 proband); congenital myopathy with predominant facial weakness with micro/retrognathia (∼10 probands); isolated or syndromic congenital facial weakness and normal eye movements (∼50 probands); and Moebius syndrome (∼250 probands). Among these, MYMK variants were identified in two individuals: an affected child from the USA who had been misdiagnosed with Moebius syndrome inherited M1 from her father and c.2T>A (p.0?, M4) from her mother (Family 4, individual 7); and an affected child from a consanguineous Brazilian pedigree diagnosed with CFZS was homozygous for a c.461T>C (p.Ile154Thr, M5) variant (Family 5, individual 8; the affected sibling was not enrolled in the study; Fig. 1a, Table 1, Supplementary Fig. 1a). Sequencing also revealed nine known polymorphisms, none of which were found in combination with other compound heterozygous changes (Supplementary Table 1d).

Among the 5 MYMK missense variants, only M1 is present in the Exome Aggregation Consortium (ExAC) database[16]; it has a cumulative MAF of 0.0013 (0.002 for Europeans) and is reported in the homozygous state in one individual with no declared phenotype who is not recontactable. We constructed and compared families' haplotypes surrounding the MYMK locus. The four families segregating M1 are haploidentical within a 41 kb region, suggesting a common origin for this allele. By contrast, the two pedigrees segregating M3 share maximum haploidentity of 1.4 kb, suggesting that this variant, present in a CpG dinucleotide and therefore prone to mutation[17], arose independently (Supplementary Fig. 1b). MYMK is predicted to have seven transmembrane (TM) domains with amino- and carboxyl-termini located extracellularly and intracellularly, respectively[18]. The five corresponding amino acid substitutions alter phylogenetically highly conserved residues and are predicted to be deleterious by SIFT and Polyphen2 (Fig. 1b,c, and Supplementary Table 1e).

**The CFZS phenotype differs from other congenital syndromes.** To define the MYMK human phenotype, we examined affected individuals and determined the penetrance of their clinical features and laboratory test results (Fig. 2, Table 1, Supplementary Fig. 2). All eight affected individuals had congenital bilateral facial weakness, upturned/broad nasal tip, micro/retrognathia, generalized muscle hypoplasia with mild axial and appendicular weakness, delayed motor milestones, and normal cognition. None had abducens nerve palsy, although mild to minimal eye movement limitations were noted in extreme positions of gaze. Both males also had cryptorchidism, while the penetrance of eleven additional clinical features was reduced (Fig. 2a, Supplementary Fig. 2a–e, Table 1). Notably, all eight individuals achieved independent mobility in childhood, and none reported significant progression of appendicular muscle weakness, required assistance with ambulation, or have shown susceptibility to malignant hyperthermia. Individuals 1 and 2, the eldest participants and siblings, both had nocturnal hypoventilation and developed restrictive pulmonary disease and pulmonary hypertension after age 35 years, from which individual 1 died at the age of 37 years. They also had a family history of idiopathic pulmonary hypertension, and individual 2 has had very slowly progressive axial weakness. Therefore, it remains to be determined if pulmonary

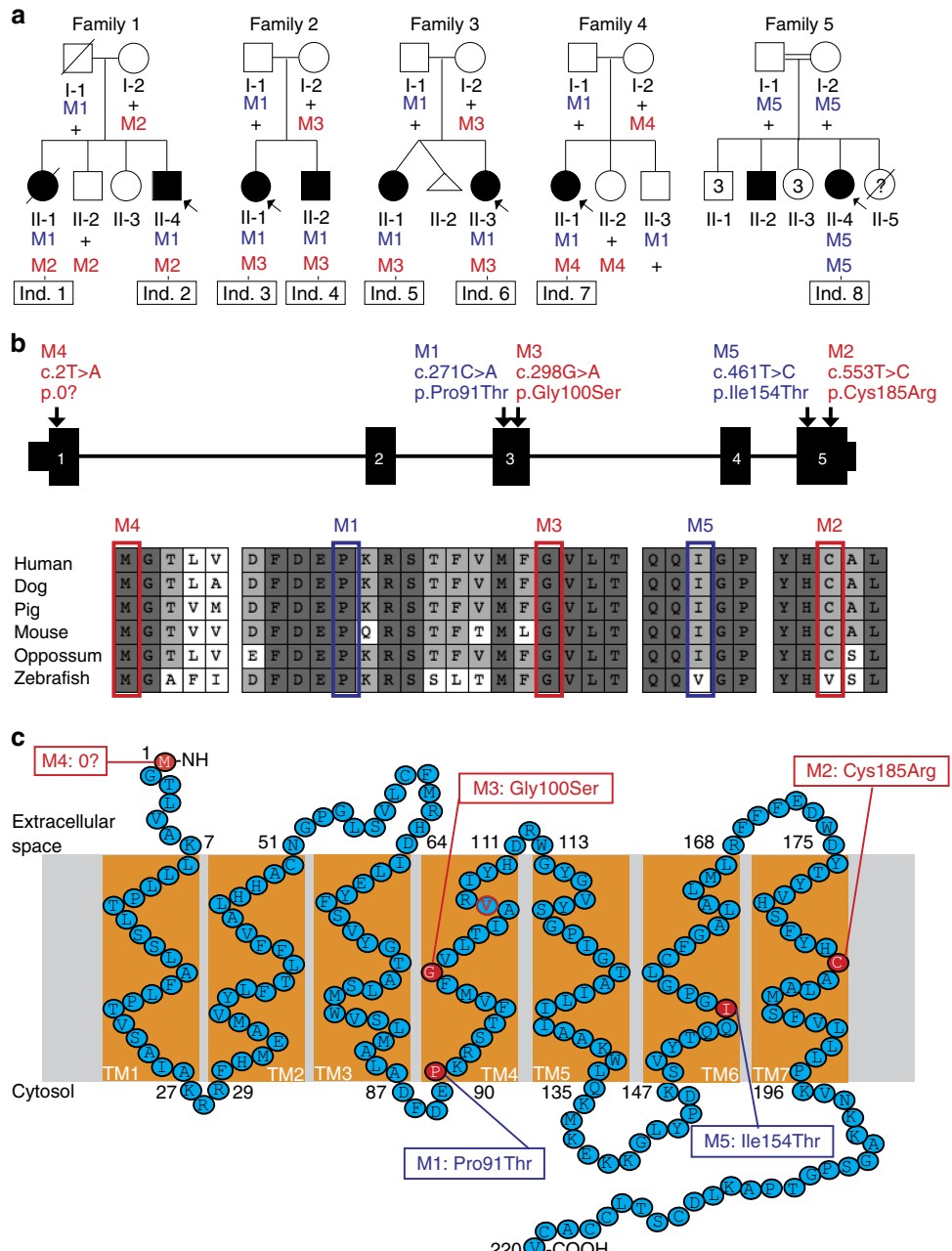

**Figure 1 | *MYMK*-CFZS genetics.** (**a**) Pedigree schematics. Filled symbol = affected; arrow = proband. Mutation status is indicated under each enrolled individual:  + = WT allele, M1–M5 = mutant alleles as defined in **b**, red and blue fonts denote null and hypomorphic alleles, respectively. Individuals' numbers (Ind. 1–8) are boxed in black. (**b**) Top: Human *MYMK* gene schematic. Arrows denote nucleotide position of M1-M5 mutations. Bottom: multispecies protein MacVector alignment of amino acids surrounding each substitution: dark grey, light grey, and white shading indicate conserved, partially conserved, and nonconserved residues, respectively. (**c**) 2D structure of the human 220 amino acid MYMK protein (previously referred to as the 7 transmembrane domain (TM) TMEM8C protein) (modified from Millay *et. al.*[18]), with starting and ending amino acid residues of each TM numbered. The red-filled residues denote the predicted locations of the CFZS amino acid substitutions. M4 is predicted to preclude initiation of wildtype translation; if translation were initiated instead by the next methionine codon (Met32), the first TM would be excluded from the protein. M2 and M3 alter residues buried deeply in TM7 and TM4, respectively. M3 alters Gly100, which is highly conserved in MYMK, TMEM8A and TMEM8B[18]. M2 alters a less conserved Cys185, but introduces a charged arginine into a hydrophobic domain. This model locates M1 and M5 at the beginning of TM4 and five residues inside TM5, respectively, and both substitute a hydrophobic amino acid with threonine. The red 'V' in residue 106 in TM4 corresponds to the amino acid disrupted by the one-base-pair frameshift-causing insertion generated in the zebrafish model (*mymk^insT*). TM, transmembrane domain; M, mutation as per (**b**) followed by corresponding amino acid substitution. See also Supplementary Fig. 1.

hypertension and/or adult-onset progressive weakness are features of *MYMK* CFZS. When obtained, brain magnetic resonance images (MRI) were normal (Table 1, Supplementary Fig. 2f). Affected individuals had no fasciculations or other clinical signs of peripheral nerve involvement, electromyogram/nerve conduction studies (EMG/NCS) revealed myopathic (and not neurogenic) changes, and *Mymk* expression was not detected in adult mouse sciatic nerve (Table 1, Supplementary Figs 2g,h and 3u),

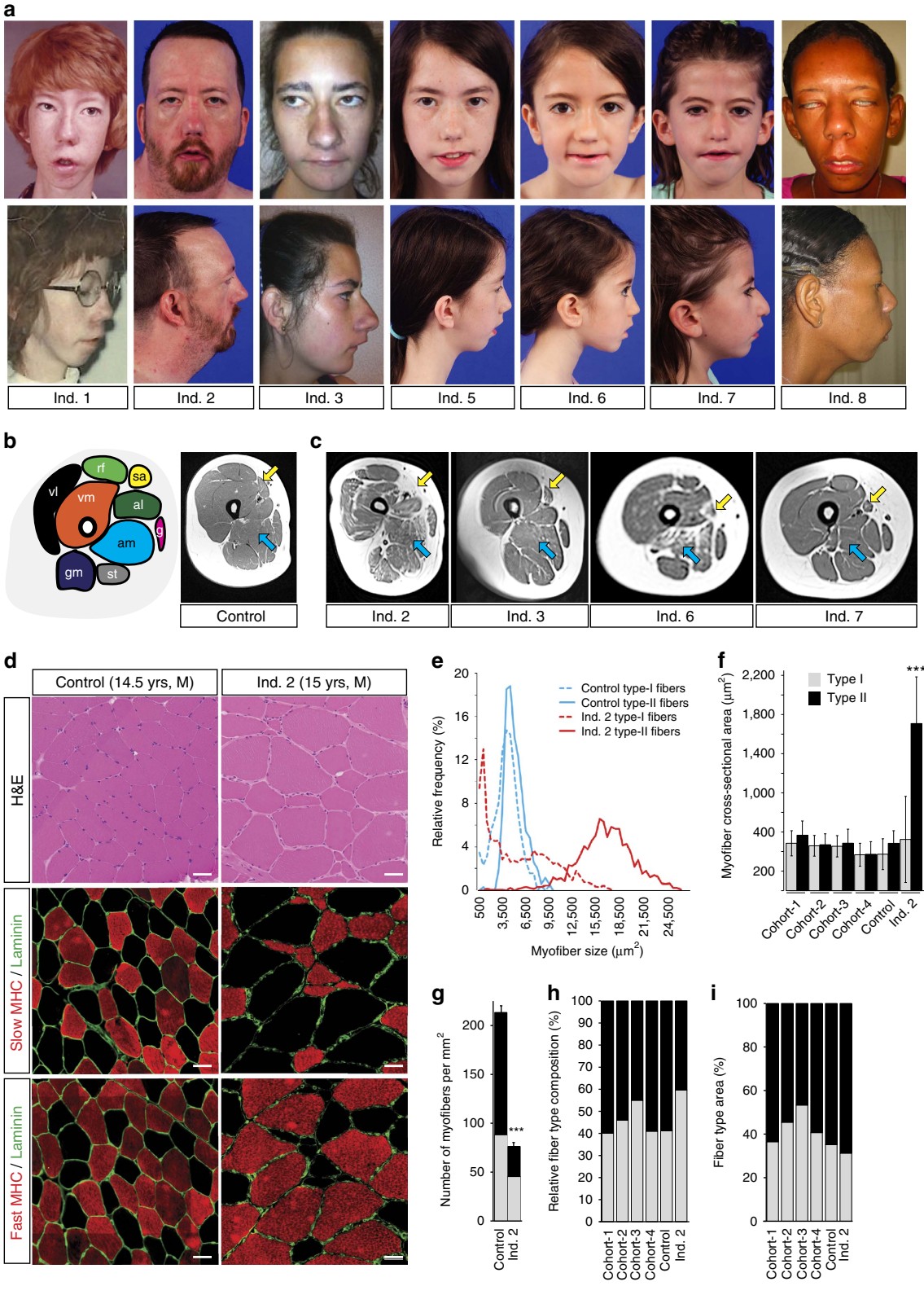

supporting the absence of a neurogenic component to CFZS. Muscle MRI revealed fatty infiltration predominantly affecting the adductor magnus and sartorius muscles, and to a lesser degree the paraspinal and glutei maximi muscles. More pronounced changes were observed in the oldest living participant (Individual 2; Fig. 2b,c, Table 1, Supplementary Fig. 2h). Scoring the presence or absence of a maximum of 19 clinical and 4 laboratory features revealed variability over the cohort with scores ranging between 68 and 96% for a given affected individual (Table 1).

Individual 2 underwent biopsy of the right vastus lateralis muscle at 15 years of age. Histologic analysis revealed a disproportionately high number (60%) of variably sized type I (slow) fibers, many of which were atrophic/hypotrophic, polygonal, and often present in small groups. In contrast, type II (fast) fibers were consistently hypertrophic with cross-sectional areas measuring 330% of control type II fibers[19–22] (Fig. 2d–i). This type I fiber type predominance and hypotrophy with type II fiber hypertrophy meets the diagnostic criteria for congenital fiber type disproportion myopathy (CFTD). Additional staining and analyses (Supplementary Fig. 3a–t) showed no rods, cores, reciprocal type II fiber grouping, or increased central or number of nuclei. The biopsy also lacked features of dystrophies, myositis, storage diseases, or mitochondrial myopathies. Notably, Individual 2 had a prior quadriceps biopsy as a toddler that was reported to show a slight reduction in fiber size and a moderate increase in perimysial connective tissue. Thus, while a few regenerating fibers with myogenin-positive nuclei were identified (Supplementary Fig. 3j,o), the appearance of the more distinctive muscle biopsy features shown here that developed later in childhood, and the greater degree of fatty infiltration of affected muscles in the older compared to the younger participants support reduced regenerative capacity.

**CFZS results from *MYMK* null or hypomorphic alleles.** Unlike $Mymk^{-/-}$ mice, CFZS patients survive and thus may retain residual *MYMK* function. Therefore, based on the combinations of genotypes harboured by the affected individuals, we hypothesized that M1 and M5 were hypomorphic and M2, M3, and M4 were null alleles (Supplementary Fig. 4). To evaluate this hypothesis, we first tested the effect of the variants on protein stability by overexpressing FLAG-tagged WT and mutant *MYMK* alleles in HeLa cells. By western blot, WT but no mutant protein of the expected band size was visible in the soluble fraction after 24 h, despite comparable transfection levels (Supplementary Fig. 5a,b), suggesting that the mutant alleles are unstable and degrade and/or aggregate, similar to other mutant membrane proteins[23]. We found, however, that immunostaining with a FLAG antibody revealed a low percentage of M1 and M5 transfected protein at the membrane compared to WT. Notably, there was no M2, M3 or M4 protein at the membrane, and transfected M2, M3, M4, and M5 proteins formed cytoplasmic aggregates (Supplementary Fig. 5c,d).

Based on these data, we next asked whether M1 and M5 fusogenic activity remained intact by exploiting the observation that ectopic overexpression of WT mouse or human *Mymk/MYMK* in fibroblasts leads to fibroblast fusion with C2C12 myoblasts[11,18]. Indeed, fibroblast overexpression of FLAG-tagged human *MYMK* M1 or M5 constructs induced fusion with C2C12 cells, supporting the hypothesis that, despite lower expression level, they retain residual function. Moreover the M1 allele that is reported as homozygous in one individual in the ExAC database had similar fusion as WT, while M5 had reduced fusion. By contrast, overexpression of M2, M3 or M4 constructs resulted in no fusogenic activity, supporting these as null alleles (Fig. 3a–c).

Primary myoblasts were available from individual 2, who is heterozygous for the M1 and M2 alleles. We observed no difference in the capability of his CFZS mutant cells to differentiate compared to control myoblasts (Fig. 3d). Following differentiation, however, there was a significant difference in the fusion index, with a higher percentage of singly-nucleated relative to multinucleated cells in CFZS compared to control myoblasts (Fig. 3e). The differentiated CFZS-derived myoblasts also appeared hypotrophic, with reduced cell diameter and elongated cytoplasm compared to the control (Fig. 3f,g). While normal variability exists among human-derived myoblasts from even a single affected individual, the reduced fusion and hypotrophy of CFZS-mutant myoblasts, while not as severe, parallels the findings in murine $Mymk^{-/-}$ myoblasts where no fusion occurs[11].

**$Mymk^{insT/insT}$ zebrafish lack fast-twitch myoblast fusion.** To confirm pathogenicity and determine if the consequences of null and hypomorphic *MYMK* alleles could be distinguished *in vivo*, we modelled CFZS variants in zebrafish. Unlike human and mouse, in which both type I slow-twitch and type II fast-twitch progenitors fuse to form multinucleated syncytia, zebrafish slow-twitch muscle progenitors do not fuse. Moreover, the relative proportions and organization of zebrafish slow- and fast-twitch fibers change over development and are spatially separated, aiding in their analysis[24]. Consistent with its role in muscle fusion, zebrafish *mymk* is exclusively expressed in multinucleated fast-twitch fibers[13], whose differentiation and fusion begins after adaxial cell migration and is completed around 24 h post fertilization (hpf)[25]. Knockdown of *mymk* in zebrafish[13] and

**Figure 2 | Clinical and radiological and pathological features of *MYMK*-CFZS.** (**a**) Front (top) and profile (bottom) facial photos of individuals (Ind.) 1–8 highlight facial weakness with flattened nasolabial folds, thin elongated face, midface hypoplasia, prominent nose, micro/retrognathia, and in some, low-set ears (also lagophthalmos on attempted lid closure, Ind. 8, top). (**b**) Map of left thigh muscles: vastus lateralis (vl), black; vastus medialis (vm), orange; rectus femoris (rf), light green; sartorius (sa), yellow; adductor longus (al), dark green; adductor magnus (am), light blue; gracilis (g), pink; semitendinosus (st), grey; gluteus maximus (gm), purple. Right: left thigh control MRI with sa and am indicated by yellow and blue arrow, respectively. (**c**) Left thigh MRIs of Ind. 2, 3, 6, and 7 at ages 37, 19, 7.5, and 7 years, respectively, reveal variable muscle involvement with disproportionate atrophy and fatty infiltration of the sa (yellow arrows) and am (blue arrows). (**d**) Compared to control quadriceps (left) Ind. 2 quadriceps biopsy (right, vastus lateralis) shows variable fiber size (top, hematoxylin and eosin (H&E)), atrophic/hypotrophic type-I fibers (middle, slow myosin heavy chain (MHC), red), hypertrophic type-II fibers (bottom, fast MHC, red). Scattered type IIc fibers (slow and fast MHC double-positive) were present in the control (shown here) and Ind. 2. Laminin (green). Scale bar, 50 μm. (**e**) Relative type-I (dashed lines) and type-II (solid lines) myofiber size frequency in control (blue), Ind. 2 (red). (**f–i**) Ind. 2 versus averaged cohorts of adolescent and young adult male controls (vastus lateralis, age 16–28, $n = 383$)[19–22] revealed: (**f**) significantly increased type II fiber cross-sectional area of $17,077 \pm 4770 \mu m^2$, which is 330% of measured ($4,842 \pm 1,288 \mu m^2$) and reported controls and corresponding type II fiber minimum Feret diameters of 185% of controls (type I/type II: $60 \pm 31/122 \pm 21 \mu m$ versus $57 \pm 16/66 \pm 11 \mu m$); (**g**) significantly reduced myofibers per $mm^2$, and (**h**) increased type-I fiber proportion (60%), but (**i**) only slightly decreased overall cross-sectional area composition of type I versus type II fibers (average fiber area × relative composition), consistent with a compensatory response. Significance of fiber size (cross-sectional area and minimum Feret diameter) and number and nuclei number were assessed with Student's *t*-tests and analysis of variance (ANOVA). Mean ± s.d.; ***$P < 0.0001$. See also Supplemental Figs 2, 3.

**Table 1 | Clinical features of CFZS individuals.**

| Individual ID | 1 | 2 | 3 | 4 | 5 | 6 | 7 | 8 | |
|---|---|---|---|---|---|---|---|---|---|
| Sex | F | M | F | M | F | F | F | F | |
| Age (years) | Died -37 | 37 | 19 | 1.3 | 11 | 7.5 | 7 | 28 | |
| Geographic location | USA | USA | NZ | NZ | USA | USA | USA | Brazil | |
| Racial origin | White | White | White | White | White | White | White | White, Indian, African | |
| Site of examination | Utah | NIH CC | Auckland | Auckland | NIH CC | NIH CC | NIH CC | Brazilia | |
| *MYMK* variant 1 | M1 | M1 | M1 | M1 | M1 | M1 | M1 | M5 | |
| *MYMK* variant 2 | M2 | M2 | M3 | M3 | M3 | M3 | M4 | M5 | |
| Phenotype | | | | | | | | | % |
| Facial weakness | + | + | + | + | + | + | + | + | 100 |
| Upturned/broad nasal tip | + | + | + | + | + | + | + | + | 100 |
| Micro/retrognathia | + | + | + | + | + | + | + | + | 100 |
| No abducens nerve palsy* | + | + | + | + | + | + | + | + | 100 |
| Normal cognition | + | + | + | + | + | + | + | + | 100 |
| Delayed motor milestones | + | + | + | + | + | + | + | +/− | 100 |
| Generalized muscle hypoplasia | + | + | + | + | + | + | + | + | 100 |
| Congenital contractures | + | + | + | + | Talipes | − | 4, 5th finger | 4, 5th finger | 87 |
| Growth failure | + | + | + | + | + | + | + | + | 87 |
| Feeding problems | + | + | + | + | + | + | + | − | 87 |
| Ptosis | + | + | + | − | + | − | + | + | 75 |
| Palate high/cleft | +/Cleft | +/ Cleft | − | − | + | + | + | + | 75 |
| Gastro/jejunostomy | − | + | + | + | + | + | − | − | 62 |
| Thin tubular neck | + | − | + | − | + | − | + | + | 62 |
| Pectoralis hypoplasia | + | + | − | − | + | − | + | − | 62 |
| Hypoglossia | + | + | − | + | − | − | + | − | 50 |
| Scoliosis | + | + | − | − | − | + | − | + | 50 |
| Pulmonary hypertension | + | + | − | − | − | − | − | − | 25 |
| Cryptorchidism (M) | NA | + | NA | + | NA | NA | NA | NA | 100 |
| Clinical score | 17/18 | 18/19 | 13/18 | 13/19 | 15/18 | 12/18 | 15/18 | 13/18 | |
| *Laboratory testing* | | | | | | | | | |
| CPK (IU/L) | N | 589 (<308) | NA | NA | 360 (<149) | 282 (<149) | N | 543 (<165) | 67 |
| EMG | Myopathic | Myopathic** | NA | NA | Myopathic | NA | NA | Myopathic | 100 |
| Muscle MRI | NA | ● | ¶ | NA | ● | ● | ● | NA | 100 |
| Brain MRI | NA | N | NA | NA | N | N | N | NA | 100 |
| Total score | 18/20 | 22/23 | 14/19 | 13/19 | 19/22 | 15/21 | 17/21 | 15/20 | |

F, female; M, male; NIH CC, National Institutes of Health Clinical Center; NA: not available; NZ, New Zealand; USA, United States of America; yrs, years.
M1: c.271C>A, p.P91T. M2: c.553T>C, p.C185R. M3: c.298G>A, p.G100S. M4: c.2T>A,p.0?. M5: c.461T>C, p.I154T. %, percentage of the affected individuals with this sign or symptom. Phenotype. +, present; −, absent; * individuals can have minimal limitations of eye movement in extreme positions of gaze; Muscle MRI: (●) fatty infiltration/hypoplasia of the paraspinal, gluteus maximus, adductor magnus and sartorius muscles in the thigh. (¶) Individual from NZ had isolated sartorius involvement. N: normal or within normal limits. **EMG of individual 2 of the right distal leg (TA, tibialis anterior) and right proximal arm (triceps) revealed short duration motor units and early recruitment, and quantitative motor unit analysis (QMUAP) had a short mean duration; EMG of the TA, triceps, and right orbicularis oris revealed no abnormal spontaneous activity.

chick[12] myoblasts were reported to produce a fusion defect similar to that in *Mymk*$^{-/-}$ mice. We recapitulated a similar muscle fusion phenotype by the injection into zebrafish of a splice-blocking and three translation-blocking morpholinos (Supplementary Fig. 6). The muscle fusion defects following morpholino knockdown were incomplete and variable, however, and thus we employed CRISPR-Cas9 technology to generate a germline loss-of-function *mymk* fish line.

We generated a mutant *mymk* zebrafish line harbouring a single thymidine insertion in exon 3 (c.434insT; p.Val106CytfsTer33; Fig. 1c). F2 *mymk*$^{insT/insT}$ embryos lacked fast-twitch myoblast fusion at 24 h.p.f. (Fig. 4a). By 48 h.p.f., the nuclei of the unfused myofibers aligned fully at the center of each myotome, in contrast to the random-appearing distribution of WT myonuclei (Fig. 4b, Supplementary Video). This is the same phenotype reported recently in two other truncating loss-of-function *mymk* zebrafish alleles[14], and is a similar but far more striking and penetrant phenotype than that observed in morpholino-treated embryos (Supplementary Fig. 6). As predicted, loss of mymk

function did not alter the organization of slow-twitch myofibers, as they do not fuse in WT teleosts (Supplementary Fig. 7d).

In contrast to *Mymk*$^{-/-}$ mice that do not survive, and consistent with the two previously reported truncating alleles[14], *mymk*$^{insT/insT}$ zebrafish were viable, likely because of the residual function of the slow-twitch mononucleated myofibers (Supplementary Fig. 7f). Larval and early juvenile *mymk*$^{insT/insT}$ fish were macroscopically indistinguishable from their heterozygous and WT siblings. By contrast, adult fish were small and had developed craniofacial deformities (Fig. 4c–e). Microscopic examination of their fast-twitch musculature revealed variably-sized hypotrophic fibers with variable degrees of fatty infiltration compared to their WT siblings (Fig. 4f). Heterozygous *mymk*$^{insT/wt}$ embryos and adults appeared normal. A similar mutant phenotype was observed in adult F0 fish injected with a guide targeting exon 5 of *mymk*, confirming that this later-onset phenotype was dependent on loss of *mymk* function.

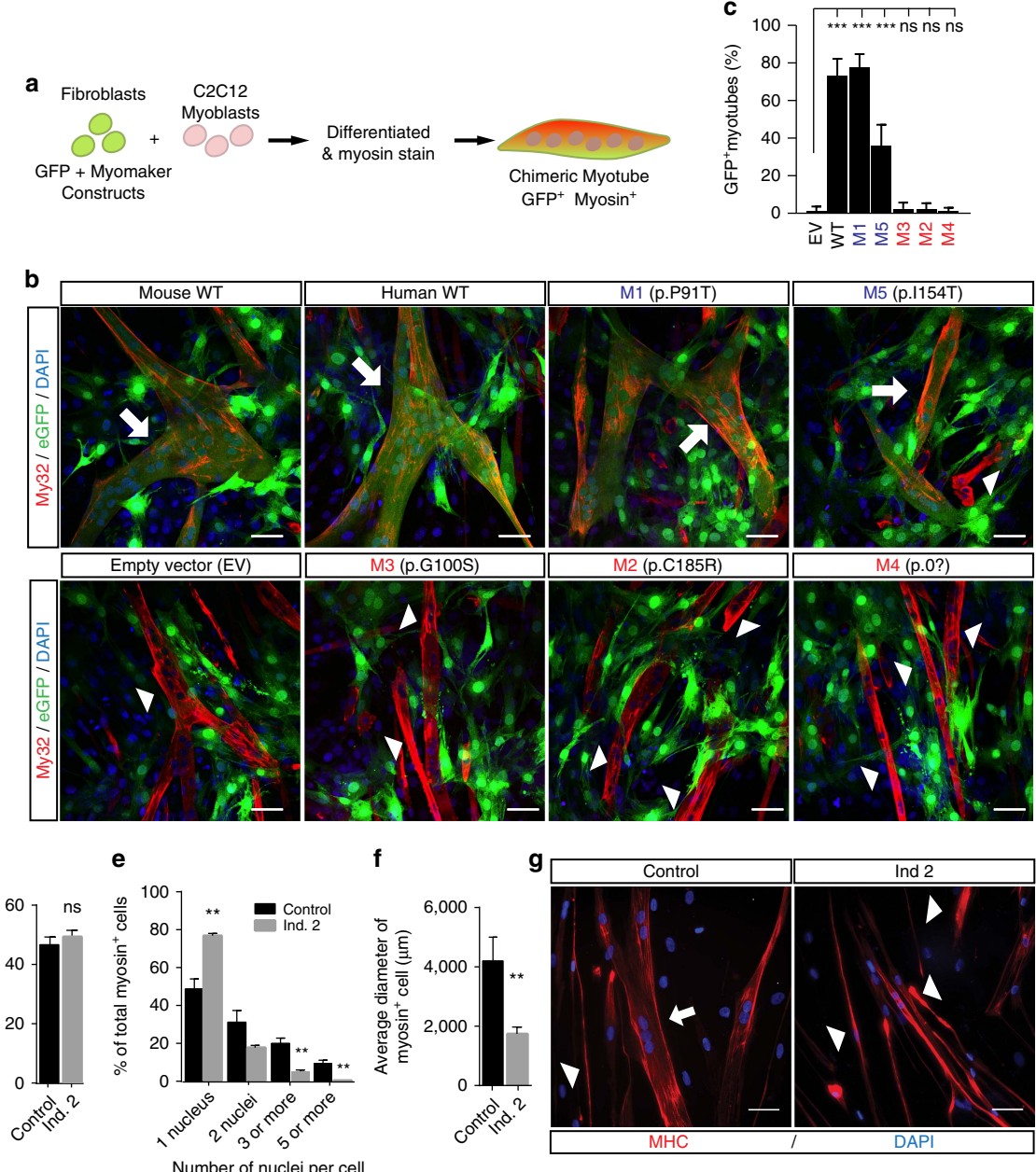

**Figure 3 | CFZ mutations cause impaired fusion in ectopic expressing cell lines and patient derived cell lines.** (**a–c**) Fibroblast-myoblast cell-cell fusion experiments: (**a**) Schematic of heterologous cell–cell fusion system. Fibroblasts were co-infected with GFP plasmid and FLAG-tagged WT or CFZS-mutant *MYMK* construct, co-cultured with C2C12 myoblasts for 4 DIV, and immunostained with myosin antibody, a myocyte marker. If fibroblast-myoblast fusion occurs, chimeric myotubes appear yellow/orange from colocalization of GFP (green, fibroblast origin) and myosin (red, myoblast origin). (**b**) Fibroblasts co-infected with mouse or human WT *MYMK-FLAG* constructs and GFP fuse with C2C12 cells. Hypomorphic CFZ mutations M1 (p. P91T) and M5 (p. I154T) retain fusogenic activity, while fibroblasts infected with empty vector or null CFZ mutations M2-M4 do not. White arrows indicate GFP⁺ myosin⁺ yellow fused myotubes, whereas white arrowheads highlight unfused myosin⁺ myocytes. (**c**) Quantification of heterologous fusion between infected fibroblasts and C2C12 cells. The number of GFP-positive myotubes was counted and normalized by the total number of Myo32 positive myotubes. (**d–g**) CFZS primary myoblasts fusion experiments: (**d**) Differentiation index, calculated as number of nuclei in myosin positive cells divided by the total number of nuclei in all cells per field, shows no difference between control and Ind. 2 primary myoblast cell lines. (**e**) Quantifications of the number of nuclei in myosin⁺ cells reveal a significant increase in single nucleated cells and decrease in multinucleated cells in Ind. 2 derived cells compared to control. (**f**) The cell diameter of Ind. 2 myoblasts is significantly reduced compared to control myoblasts ($n = 45$). (**g**) Image of control myoblasts (left) reveals greater width, more cytoplasm, and more nuclei per fiber (white arrow) than Ind. 2 myoblasts (right—white arrowhead) at 7 DIV post differentiation. Blue: DAPI stain of nuclei. Red: myosin heavy chain A4.1025 antibody. Analyses of fusion assay and differentiated myoblasts were each performed on at least 6 and 10 random fields per experiment, respectively, and 3 independent experiments. Statistical analyses in: (**c**) calculated by one way ANOVA with Bonferroni correction for multiple testing and shown as comparison to the empty vector (EV); (**d–f**) using two-tailed Student's unpaired *t* test. Mean ± s.e.m.; \*\**P* < 0.001; ns, not significant. Scale bars in **b**,**g**, 50 μm. See also Supplementary Fig. 5.

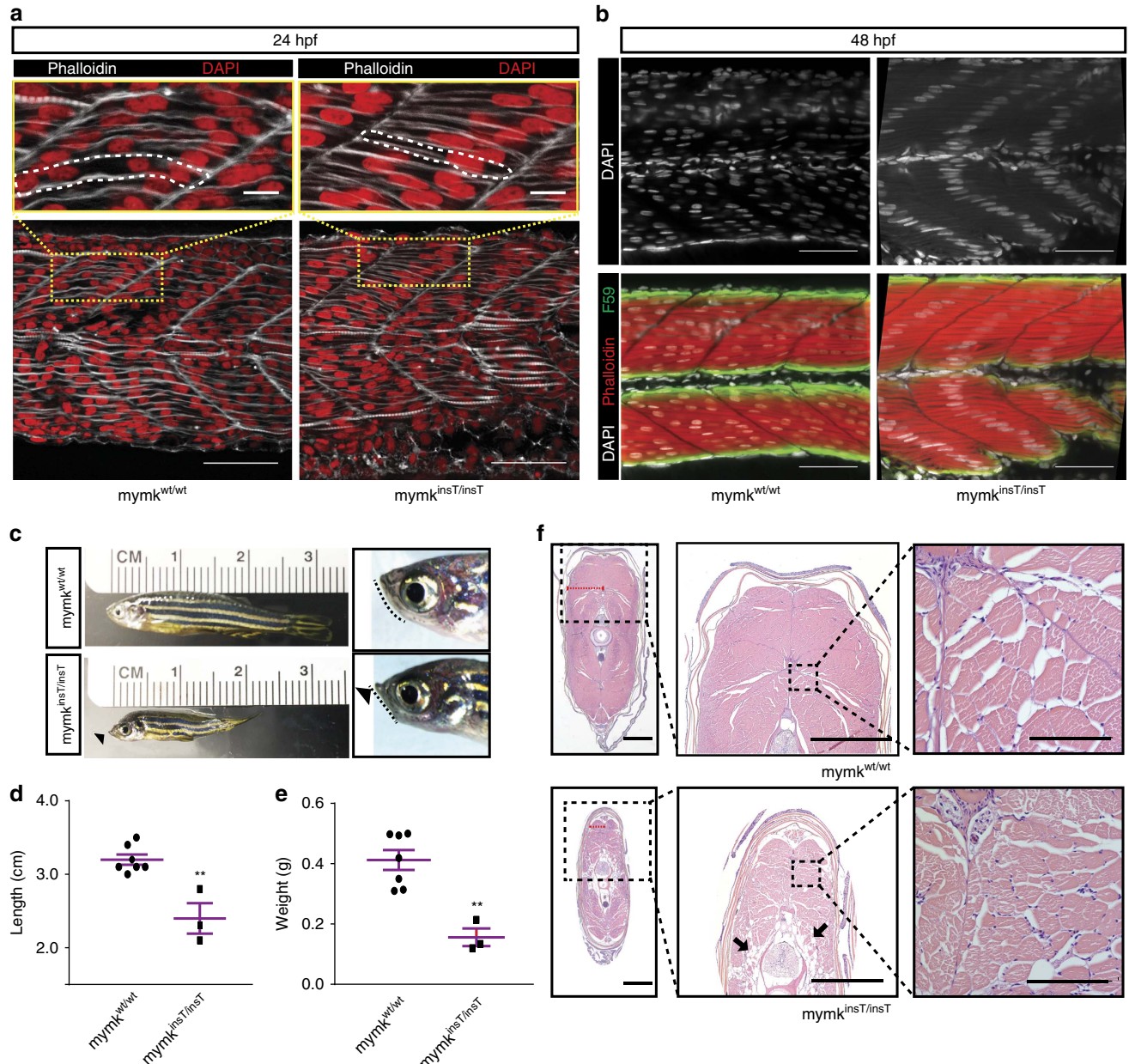

**Figure 4 | *mymk^insT/insT* zebrafish embryos lack fast-twitch myoblast fusion and adults have CFZS-like myopathic features.** (**a**) 24 h.p.f. embryos stained with 488-conjugated phalloidin (white) and DAPI (red) to label the forming myofibers and nuclei, respectively. WT *mymk^wt/wt* embryo myofibers are multinucleated, while *mymk^insT/insT* embryo myofibers elongate and differentiate, but fail to fuse. (**b**) Upper panel: 48 h.p.f. embryos stained with DAPI show the distribution of myonuclei in fused mymk^wt/wt (left) and unfused (right) *mymk^insT/insT* fast-twitch myofibers. Compared to the random appearance of WT myonuclei, the single nuclei in unfused *mymk^insT/insT* myofibers pathologically align at the center, equidistant between two myosepta. Lower panel: Merged images showing phalloidin (red), DAPI (white) and F59 (green, anti-myosin heavy chain) confirm that affected myofibers are fast-twitch fibers because they do not express F59, a slow-twitch cellular marker in zebrafish. (**c**) A 6-month-old male *mymk^wt/wt* zebrafish (top) compared to an age- and sex-matched *tmem8c^insT/insT* fish (bottom). Male and female mutant zebrafish are small and have a flattened/retrognathic jaw (right, indicated by dotted line and black arrowhead) not appreciated during larval and early juvenile stages. By 3 months of age, jaw weakness prohibits *mymk^insT/insT* zebrafish from fully closing their mouths. (**d,e**) Adult *mymk^insT/insT* zebrafish (n = 3) are significantly shorter (**d**) and weigh less (**e**) than age and sex-matched WT siblings (n = 7). (**f**) Hematoxylin-Eosin (H&E) staining of caudal transverse sections of WT (top) and *mymk^isnT/insT* (bottom) of 6-month-old male zebrafish siblings at three magnifications. Zebrafish *mymk*-expressing fast-twitch myofibers are located centrally, while *mymk*-negative slow-twitch myofibers are located near the body wall and stain slightly darker with H&E. The mutant fish have reduced body width (compare the red line extending from the dorsal artery to the body wall in WT versus mutant fish in left photos), and fat infiltration (thick black arrows, middle photo) that is absent in the WT fish. Fast-twitch myofibers appear smaller compared to WT (right). Statistics by two-tailed Student's unpaired *t*-test; mean ± s.e.m.; **P < 0.002. Scale bars, (**a,b**) main images 50 μm, insets 10 μm. (**f**) left and center panels 500 μm, right panels 50 μm. See also Supplementary Figs 6,7 and Supplementary Video.

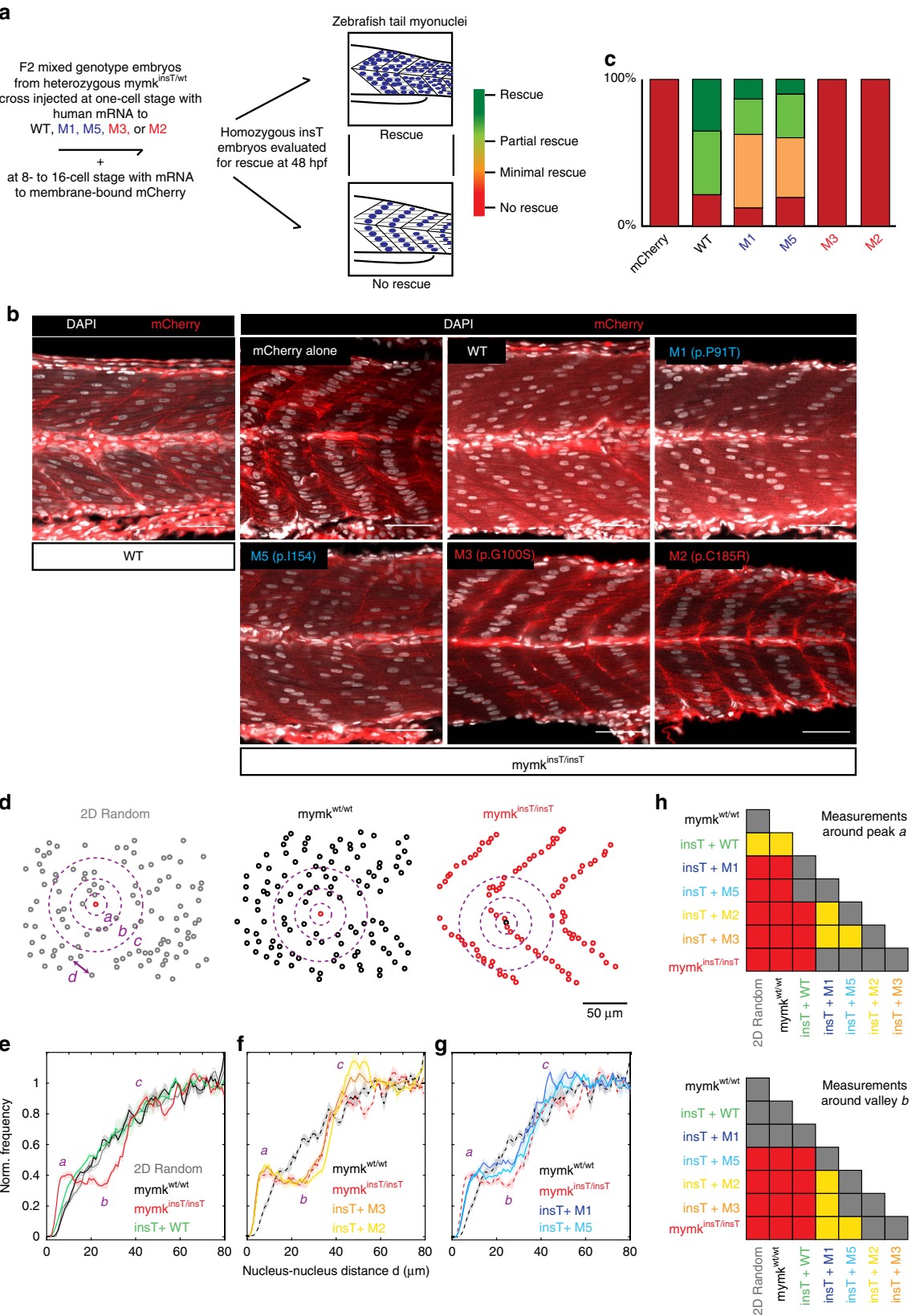

**WT or hypomorphic *MYMK* mRNA rescues *mymk^insT/insT* fish.**
We used the *mymk^insT/insT* loss-of-function zebrafish line to confirm the pathogenicity of the CFZS alleles *in vivo* and their function as null or hypomorphic alleles. We injected WT or mutant human *MYMK* mRNA and membrane bound mCherry fluorescent protein[26] into *mymk^insT/insT* embryos, and visualized myonuclear alignment at 48 hpf to evaluate the ability of the injected mRNA to rescue the myoblast fusion defect (Fig. 5a). By blinded qualitative analysis, injection of WT *MYMK* mRNA resulted in myonuclei distributed in a random- or near-random-appearing pattern in 80% of embryos, thus rescuing the *mymk^insT/insT* phenotype. By contrast, injection of M2 or M3 *MYMK* mRNA resulted in no rescue of the *mymk^insT/insT* phenotype, and was indistinguishable from injection of mCherry alone, while injection of M1 or M5 *MYMK* mRNA resulted in an intermediate level of rescue (Fig. 5a–c). We obtained similar but less penetrant results when we co-injected WT or mutant mRNA together with a transcription-blocking morpholino into WT embryos (Supplementary Fig. 6d,f,g).

To quantify the rescue phenotypes, we measured the distances between myonuclei in *mymk^wt/wt* embryos, *mymk^insT/insT* embryos, and *mymk^insT/insT* embryos injected with WT-, M1-, M2-, M3-, or M5- *MYMK* human mRNA. For each injected allele, we then calculated for each nucleus the frequency distribution of all nuclei within a 5 µm to 80 µm distance (Fig. 5d). *mymk^wt/wt* embryos, and *mymk^insT/insT* embryos injected with WT mRNA, had a random distribution of myonuclei, while *mymk^insT/insT* did not (Fig. 5d). The distribution of nuclei in *mymk^insT/insT* embryos injected with M2- or M3-mutant mRNA were not significantly different than *mymk^insT/insT* alone, confirming these as null alleles (Fig. 5d–h). By contrast, *mymk^insT/insT* embryos injected with M1 or M5 resulted in distributions intermediate between *mymk^wt/wt* and *mymk^insT/insT*, consistent with hypomorphic function (Fig. 5g,h). Moreover, while not significantly different from one another, the distribution following injection of M1-mRNA was more similar to *mymk^wt/wt*, while the distribution following injection of M5-mRNA was more similar to *mymk^insT/insT* injected with M2- or M3-mRNA. These data support the pathogenicity of the *MYMK* alleles and their null or hypomorphic status. They also suggest greater residual mymk activity of M1 over the other disease alleles, consistent with the *in vitro* fusion assay results and with the presence of homozygous M1 in the ExAC database in one individual with no declared phenotype.

**Discussion**
We demonstrate that recessive mutations in *MYMK* cause CFZS through a combination of one hypomorphic and one null allele, or two hypomorphic alleles, which reduce MYMK function and myoblast fusion below a threshold but not to zero (Supplementary Fig. 4). Despite the presence of the hypomorphic M1 allele at a very low frequency in the ExAC database in the heterozygous state and in one individual of unknown clinical status in the homozygous state, both our *in vitro* and *in vivo* data support its pathogenicity. It does, however, have the highest level of residual MYMK activity among the mutant alleles. Thus, it will be of interest to identify and phenotype individuals who are homozygous for the M1 or compound heterozygous for the M1 and M5 alleles to define the threshold of MYMK activity below which one manifests the CFZS phenotype. We also anticipate that, similar to *Mymk^−/−* mice[11], two null alleles will result in absence of muscle and be incompatible with human life (Supplementary Fig. 4).

The combined clinical, imaging, and muscle biopsy findings in CFZS define a distinct disorder. Significant facial weakness and Robin sequence, minimal limitation of horizontal end-gaze, only mild axial and appendicular weakness, and CFTD on muscle biopsy distinguishes CFZS from most other congenital myopathies. CFZS may have greatest clinical overlap with Native American myopathy resulting from *STAC3* mutations[27,28], although the absence of malignant hyperthermia in CFZS and possibly different muscle pathologies distinguish them. The characteristic distribution of fatty infiltration/atrophy on muscle imaging in CFZS is similar to that reported in *SEPN1* CFTD[29], in which respiratory muscle weakness, scoliosis, and impaired muscle regeneration are present, but facial weakness is not prominent[30,31]. CFZS pathophysiology may most closely resemble EMARDD (early onset myopathy, areflexia, respiratory distress, and dysphagia), which results from *MEGF10* mutations that appear to reduce myocyte proliferation, differentiation, and fusion[32–34]. By contrast, the ability to almost fully abduct the eyes in CFZS permits its distinction from Moebius syndrome, in which abduction limitation is complete or nearly complete[8]. Finally, as anticipated[2,6], normal cognition, absence of seizures and abducens nerve palsy, and normal brain and brainstem imaging shared by the MYMK-CFZS individuals in this report suggest that many cases previously reported to be CFZS likely represent different disease entities[2–7].

While the current CFZ muscle pathology fits best within the category of CFTD[35,36], the type II hypertrophy is greater than typically reported[37,38]. Our data also support the appearance of more distinctive muscle pathology and an increase in fatty infiltration of specific muscle groups over time, potentially correlating with slow progression of weakness and adult-onset restrictive pulmonary disease. Thus, additional longitudinal clinical data coupled with analyses of serial biopsies that target

**Figure 5 | Ectopic expression of *MYMK*-WT and -hypomorphic alleles can fully or partially rescue the *mymk^insT/insT* fusion phenotype.** (a) Approach to evaluate rescue of *mymk^insT/insT* 48 h.p.f. embryo fusion phenotype following co-injection of WT or mutant human *MYMK* mRNA. Nuclear dispersion is qualitatively stated to be low (no rescue), minimal (slight rescue), moderate (partial rescue), or high (rescue). (b) Optical sections of laterally mounted 48 hpf *mymk^wt/wt* embryo injected with mCherry, and six *mymk^insT/insT* embryos injected as per a. WT-, M1-, M5-*MYMK* mRNA injected embryos show partial rescue. M2-, M3-, and mCherry alone do not. Red: membrane bound mCherry. White: DAPI. Scale bar = 50 µm. (c) Qualitative analysis of nuclear organization of embryos in b according to colour key in a. (d) Digitized images of the distribution of randomly generated non-overlapping *xy* positions (left), *mymk^wt/wt* myonuclei (middle), *mymk^insT/insT* myonuclei (right); left-right arrow *d* denotes distance between two nuclei; circles *a*, *b*, *c* are ~10, 25, 50 µm from center nucleus, respectively. (e) Frequency of distances from each nucleus to all others. 2D random (grey), *mymk^wt/wt* (black) and *mymk^insT/insT* + WT mRNA (green) have the same trend, while *mymk^insT/insT* (red) exhibits differences at distances *a*, *b*, *c* (circles in d). Increased frequency at distance *a* reflects the increased number of near neighbors. Decreased frequency at distance *b* reflects the lack of nuclei between the rows of aligned nuclei in *mymk^insT/insT* embryos. (f) *mymk^insT/insT* embryos injected with mutant *MYMK* mRNA M2 (yellow) or M3 (orange) have normalized frequencies most similar to *mymk^insT/insT* (red dashed line) versus *mymk^wt/wt* (black dashed line), consistent with lack of rescue. (g) *mymk^insT/insT* embryos injected with mutant *MYMK* mRNA M5 (turquoise) or M1 (blue) have normalized frequencies that fall between *mymk^insT/insT* (red dashed line) versus *mymk^wt/wt* (black dashed line), consistent with partial rescue. Shaded bands = ± SEM. (h) Two sample analyses for statistical significance (Kolmogorov-Smirnov) between couples of distributions in regions *a*, *b*. Top: Statistics for nuclei that fall in a distance range of 5 µm < d < 10 µm (circle *a*). Bottom: Statistics for nuclei that fall in a distance range of 18 µm < d < 28 µm (circle *b*). Grey box: NS ($P_{KS} > 0.05$); Yellow box: *($0.01 < P_{KS} < 0.05$); Red box: **($P_{KS} < 0.01$). See also Supplementary Figs 6,7.

more significantly and specifically affected muscle groups may lead to the recategorization of this disorder.

The congenital facial weakness and Robin sequence of CFZS highlight the selective vulnerability of specific muscle groups to reduced levels of *MYMK*, and the interplay between muscle and bone development. Compared to other skeletal muscles, craniofacial muscles use unique developmental pathways[39] and have distinct molecular compositions and combinations of fiber types[40] that may, in part, account for their vulnerability. Muscle is known to influence bone development and, like CFZS, other congenital myopathies with significant facial weakness are often accompanied by secondary retrognathia, palatal defects, and facial dysmorphisms[1,41]. Similarly, $Myf5^{-/-}:MyoD^{-/-}$ mice are born without muscles, including jaw and tongue muscles, and have secondary retrognathia and cleft palate[42].

While complete loss of *Mymk* in mice is perinatal lethal, zebrafish live and phenotypes only become evident at a late juvenile age. Notably, the relative number and distribution of fast- and slow-twitch muscle fibers in zebrafish change during maturation, and these fiber type changes are believed to be necessary for optimization of swimming ability and feeding behaviours as the fish grow and mature from larvae into adults[43,44]. Specifically in cranial musculature, including in the adductor mandibulae muscle, there is a gradual increase in the proportion of fast-twitch fibers during larval and juvenile stages, followed by the preferential growth and expansion of fast-twitch muscle beds during the juvenile to adult transition[43]. Fast-twitch but not slow-twitch myoblasts fuse in zebrafish, and *mymk* is expressed only in fast-twitch myoblasts. Thus, failure of fusion of the fast-twitch fibers during this normal fiber type adaptation, coupled with reduced regenerative capacity suggested by progressive fatty infiltration of muscle, may explain the onset of the $mymk^{insT/insT}$ phenotype in the late juvenile stage, and the appearance of micrognathia in adult zebrafish.

While both type I and type II human striated muscle fibers fuse during muscle development, it remains to be determined if different muscles and/or different fiber types require different levels of fusion and/or express different levels of *MYMK*, resulting in selective vulnerability to a reduction in functional MYMK protein. Future studies should both clarify this selective vulnerability and more clearly define the CFZS clinical course. Finally, the ability to genetically rescue the mutant phenotype in zebrafish lends hope for future therapies that restore *MYMK* function in muscle and lessen any potentially progressive features of this disorder.

## Methods

**Subjects.** Research participants were enrolled under protocols approved by the Institutional Review Boards of the appropriate participating Institutions: University of Utah, Salt Lake City, UT; University of Otago, Dunedin, New Zealand; Boston Children's Hospital, Boston, MA; National Human Genome Research Institute, National Institutes of Health, Bethesda, MD; Johns Hopkins Medicine, Baltimore, MD; Icahn School of Medicine at Mount Sinai, New York, NY. Studies were performed in compliance with US 45.CFR.46. Adult participants and guardians of children provided written informed consent for participation and for publication of identifying photographs. Four affected individuals and their carrier parent(s) were re-examined at the NIH Clinical Research Center under protocol 14-HG-0055 'Study on Moebius syndrome and other congenital facial weakness disorders' (ClinicalTrials.gov identifier: NCT02055248). Authors and referring physicians submitted clinical data for the other four affected individuals.

**Exome sequencing and analysis.** *Exome analysis of family 1.* Whole exome sequencing was carried out on blood DNA samples isolated from the proband, affected sister, unaffected brother and parents. Exome capture and DNA sequencing were performed at the University of Utah High Throughput Genomics Core Facility using the Agilent SureSelect Human Exon V5 kit (Agilent Technologies Inc.) and the Illumina HiSeq DNA sequencing system. Paired-end raw sequencing data from each individual were aligned to the reference human

genome (GRCh37) with two alignment programs, BWA[45] and Novoalign (www.novocraft.com). Using the software package SAMtools[46], the aligned sequencing reads were converted and merged to sorted and indexed BAM files. The standard GATK pipeline (https://www.broadinstitute.org/gatk/) was then used to call sequence variants. Functional annotation and filtering of sequence variants were performed with ANNOVAR[47] and with the DNA-Seq analysis module of Golden Helix SVS (Golden Helix Inc.). Variants were classified with respect to location (exonic, intronic, splice site, 5′-UTR, 3′-UTR, upstream, downstream, or intergenic), exonic functions (nonsynonymous, synonymous, stop, frameshift insertion/deletion, etc.), minor allele frequencies (MAF) reported in various public databases (http://www.1000genomes.org/, http://evs.gs.washington.edu/EVS/, and http://exac.broadinstitute.org/), and the dbNSFP functional prediction of nonsynonymous amino acid changes (https://sites.google.com/site/jpopgen/dbNSFP). Segregation of rare (MAF < 0.01) functional candidate variants was analysed in the family using recessive and compound heterozygosity inheritance models.

*Exome analysis of family 2.* Whole exome sequencing was carried out on blood DNA samples isolated from both affected sibs and their parents. Exome capture and DNA sequencing was performed by New Zealand Genomics Ltd using the Illumina Nextera Rapid Capture Exome Kit and Illumina HiSeq 2500 DNA sequencing platform. Paired-end raw sequencing data for each individual were aligned to the reference human genome (GRCh37) with BWA and further processed according to GATK Best Practice Guidelines with HaplotypeCaller as the variant caller. Gene context annotation and dbNSFP data was added using SnpEff[48] and allele frequency information from 1000 genomes, ESP6500 and ExAC using GATK AnnotateVariants. Filtering was undertaken using SnpSift and GATK SelectVariants. Variants were classified and segregation analysed as described for Family 1.

*Exome analysis of family 3.* DNA from the proband was extracted from a blood sample using the Puregene kit (Qiagen, Valencia, CA) while DNA from the remaining family members was extracted from saliva samples using purifier solution (DNA Genotek, Canada). DNA quality was ascertained and quantified using Tapestation (Agilent Technologies, Santa Clara, CA) and 3 μg of high quality DNA were used for exome capture and sequencing. Exome capture was performed at the Ocular Genomics Institute (OGI) at Massachusetts Eye and Ear Institute using Agilent SureSelect Human Exome V4 for the proband and Agilent V5 + UTR kit for the affected sibling and parents. Captured products were sequenced using one third of a lane of Illumina HiSeq 2000 Next-Generation Sequencing system using v2.5 SBS chemistry; average flow cell lane cluster densities were $\sim 800\,K\,mm^{-2}$ and 94% average $10 \times$ coverage of the target exome. Raw reads were aligned using Best Practice pipeline developed at the Broad Institute and called variants were annotated and filtered using the xBrowse online tool (http://atgu.mgh.harvard.edu/xbrowse). Variants here harmonized with an internal database of 375 control exome sequences generated from sequencing data obtained by the Analytical and Translational Genetic Unit at Mass General Hospital. Finally, variants were filtered by passing the PASS quality filter present in GATK. Remaining variants were then classified and segregation analysed as described for Family 1.

**MYMK sequence analysis of research cohorts.** DNA from > 300 participants diagnosed with CFZS, Moebius syndrome, or other forms of syndromic congenital facial weakness enrolled at Boston Children's Hospital, Icahn School of Medicine at Mount Sinai, Baylor-Hopkins Center for Mendelian Genomics, or the NIH Clinical Research Center were Sanger sequenced for all coding exons and intron-exon boundaries of *MYMK* using ABI 377 DNA sequencer (BCH IDDRC Molecular Genetics Core Facility), ABI 3730XL DNA analyser (Genewiz, Inc. and ACGT, Inc.), or ABI 3730 (Applied Biosystems, Life Technologies, Carlsbad, Calif., USA). Primer sequences are reported in Supplementary Table 3. Sequence data were analysed and compared with normal control and reference sequence (GRCh37/hg19) for variants using Sequencher (Gene Codes Corporation), Mutation Surveyor (SoftGenetics), and/or CodonCode Aligner 3.6.1. If a mutation were identified in the proband, participating family members were screened for its presence or absence.

**Haplotype construction.** Highly heterozygous SNPs within an 80 kb region surrounding *MYMK* and dinucleotide microsatellites markers (D9S2135 and D9S1793) proximal to the gene identified in UCSC Genome Browser (https://genome.ucsc.edu), and an unannotated dinucleotide sequence (MymkMicro) in *MYMK* intronic sequence, were screened by direct Sanger sequencing in all relevant affected individuals and their parents. Primers sequences are reported in Supplementary Table 3.

**Electromyogram/nerve conduction studies.** Needle EMG studies of right distal leg (tibialis anterior), right proximal arm (triceps branchii) and perioral muscles (orbicularis ori) were performed using a concentric needle with spontaneous and motor unit activity recorded according to standard methodology. Nerve conduction studies were performed primarily on the median and sural sensory nerves and the median and peroneal motor nerves. Measurements used standard methodology

with a Nicolet Viking Select machine (Cardinal Health, Dublin, OH) and were compared to department-based normative values[49].

**Brain magnetic resonance imaging (MRI).** Brain MRI was performed on a 3T Achieva Philips scanner with 8-channel head coil at the National Institutes of Health Clinical Radiology. Multiple MRI sequences were acquired: T1 weighted $(0.94 \times 0.94 \times 1 \, mm)$, T2 weighted $(0.86 \times 0.86 \times 3.2 \, mm)$, axial fluid attenuated inversion recovery (FLAIR) $(0.86 \times 0.86 \times 3.2 \, mm)$, and high resolution brainstem images $(0.31 \times 0.31 \times 0.4 \, mm)$.

**Muscle magnetic resonance imaging.** Muscle magnetic resonance imaging (MRI) was performed using conventional T1 weighted spin echo on a 1.5-T Aera Siemens or Achieva Phillips, or 3.0-T Verio Siemens system. Non-contrast images were obtained in the axial plane of the pelvis, thighs and lower legs. Slices were 7–8 mm thick and the gap between slices varied from 6 to 10 mm. Scans were assessed for abnormal muscle bulk and for abnormal signal intensity within the different muscles.

**Muscle biopsies.** Frozen muscle tissue from a right quadriceps (vastus lateralis) muscle biopsy from individual 2 (15 years old at the time of biopsy; obtained for clinical indications) was sent to the Engle Laboratory at Boston Children's Hospital and re-evaluated with new sections and stains. For comparison, a total of 184 de-identified muscle biopsy cases at Boston Children's Hospital were reviewed for diagnoses of minimal abnormalities, and similar age and sex to individual 2. From these, nine cases were cut, stained, imaged and compared. Section thickness in Fig. 2d is 10 μm (all H&E and control MHC) and 12 μm (Ind. 2 MHC). Quantification was performed on a representative case that appeared to most closely match published findings for the vastus lateralis muscle at this age ('control' quadriceps biopsy from an active 14.5-year-old male, diagnosed as 'skeletal muscle with rare atrophic fibers,' without other diagnostic abnormalities). For nuclei counts, a second quadriceps biopsy from a 13-year-old male diagnosed as 'skeletal muscle with mild variation in fiber size' was quantified as 'Control 2'). To compare CFZS findings with a broader cross-section of individuals, we evaluated published data of vastus lateralis measurements (relative fiber type composition, fiber type cross-sectional area, and percentage fiber type areas) from four cohorts of young men (age 16–28, $n = 383$ individuals) from the literature: cohort-1 ($n = 95$, $21.1 \pm 2.4$ years)[19], cohort-2 ($n = 215$, $24 \pm 4$ years)[20], cohort-3 ($n = 55$, $16 \pm 0$ years)[21], and cohort-4 ($n = 18$, $17 \pm 1$ years)[22].

The myofiber (fiber) cross-sectional area was calculated from the entire biopsy from each individual, including 3569 fibers from individual 2, and 609 fibers from the control individual. No blinding was possible due to the characteristic phenotype of the patient muscle biopsy. Fiber number and nuclei number were measured from three distinct regions from each biopsy, normalized to number per $mm^{-2}$, based on the areas measured, and the means and standard deviations were calculated in Microsoft Excel. Relative fiber type composition (type I fiber number/total fiber number) and percentage fiber type area (type I relative fiber type composition × mean type I area/(type I relative fiber type composition × mean type I area + type II relative fiber type composition × mean type II area)) were calculated for each area of the biopsy. Image alignments were performed in Adobe Photoshop. Fiber cross-sectional area and minimum Feret diameter were measured using Image J (1 pixel in raw images = 0.3467 μm). Fibers were binned in 500 μm² increments for histogram generation. For comparative cohort 4 (ref. 22), approximate fiber area was calculated as the square of the reported diameters (which were the mean of the maximum and minimum Feret diameters). Type II A and type II B fiber sizes were combined into a single average Type II fiber size for each cohort. All other measurements from cohorts 1–4 are presented directly as reported[19-22]. Significance of fiber size (cross-sectional area and minimum Feret diameter) and number and nuclei number were assessed with Student's $t$-tests and analysis of variance (ANOVA) in Microsoft Excel. Graphs for the muscle biopsy analysis were generated in Microsoft Excel with mean ± s.d. reported. A $P$-value < 0.05 was considered significant.

**Human primary myoblast cultures.** A needle biopsy was obtained from the rectus femoris muscle of individual 2 at 37 years of age using a percutaneous needle under NIH protocol 00-N-0043 'Clinical and Molecular Manifestations of Inherited Neurological Disorders' (ClinicalTrials.gov identifier: NCT00004568). Control primary myoblasts were obtained from a hamstring open biopsy of a 30-year-old healthy white non-Hispanic female and generously provided by Louis Kunkel (Boston Children's Hospital IRB protocol #03-12-205R). Primary myoblasts were isolated from biopsy explants using the EuroBioBank protocol. In brief, muscle explants were isolated from adjacent connective and fatty tissue using a dissecting scope. Muscle explants were then plated on T25 flasks and allowed to attach. Small amounts of proliferating medium (DMEM supplemented with 20% FBS, L-glutamine, 1 × penicillin-streptomycin, insulin (10 μg ml$^{-1}$), human basic fibroblast growth factor (25 ng ml$^{-1}$)) were added and the myoblasts were allowed to grow and proliferate for several days before they were enzymatically detached and plated separately.

For culture and differentiation, primary myoblasts were grown on 0.1% gelatin-coated dishes in Skeletal Muscle Cell Growth Media with supplement (PromoCell

GmbH), complemented with 20% Fetal Bovine Serum (FBS), 1 × Antibiotic-Antimycotic (ThermoFisher), 1x Glutamax (ThermoFisher). Cells were passaged every 48 h. For differentiation into multinucleated myotubes, growth medium was switched to Skeletal Muscle Cell Differentiation Media with supplements (PromoCell GmbH), complemented with 2% Horse Serum (HS), 1x Glutamax and 1x Antibiotic-Antimycotic. Cell culture medium was refreshed every three days.

**C2C12 and HeLa and fibroblast cell cultures.** C2C12 cells, 10T1/2 fibroblasts, and HeLa cells were purchased from American Type Culture Collection and maintained in DMEM (Sigma) containing 10% heat-inactivated FBS supplemented with antibiotics. Cells were not re-authenticated and were not tested for mycoplasma. C2C12 cells were differentiated by switching to media containing 2% heat-inactivated horse serum and antibiotics. All solutions described as a percentage are based on vol/vol.

**Plasmids.** Human *MYMK* sequence cloned in a pCMV6-Entry vector was purchased from Origene and mutagenized using QuickChange II XL Site-directed mutagenesis kit (Agilent Technologies) according to the manufacturer's specifications and using the following primers:

P91T_F: 5′-CTGGCCGACTTCGACGAAACCAAGAGGTCAACATTTG-3′, P91T_R: 5′-CAAATGTTGACCTCTTGGTTTCGTCGAAGTCGGCCAG-3′.

I154T_F:5′- CCGGGGCCTGTCTGCTGGGTGTAGACG-3′, I154T_R: 5′-CGTCTACACCCAGCAGACAGGCCCCGG-3′.

G100S_F: 5′-CATTTGTGATGTTCAGCGTCCTGACCATTG-3′, G100S_R: 5′-CAATGGTCAGGACGCTGAACATCACAAATG-3′.

C185R_F: 5′-CACAGCTTCTACCACCGTGCCCTGGCTATG-3′, C185R_R: 5′-CATAGCCAGGGCACGGTGGTAGAAGCTGTG-3′.

Quality and accuracy of the mutagenesis were evaluated by direct capillary sequencing using primers spanning the T7 promoter. WT human MYMK-FLAG plasmid and human *MYMK*-FLAG constructs harbouring the individuals' mutations were designed and purchased as gBlocks gene fragment (Integrated DNA Technologies) and cloned into pBabe retroviral vector using EcoRI restriction site. The FLAG nucleotide sequence was 5′-GATTACAAGGATGACGACG ATAAG-3′, and the signal sequence upstream of FLAG was 5′-ATGAAGAC GATCATCGCCCTGAGCTACATCTTCTGCCTGGTGTTCGCC-3′. These FLAG-tagged sequences were then subcloned in a pCDNA v3.1 vector for expression in HeLa cells. For rescue of *mymk*$^{insT/insT}$ zebrafish, WT human *MYMK* and the mutant plasmids generated as described above were subcloned without the FLAG sequence in a pCS2+ backbone using *Xho*I and *Eco*R1 restriction enzymes.

**HeLa cell transfections.** HeLa cells were plated on 6-wells plate for protein extraction or on 0.1% gelatin-coated glass slides in 24-wells plate for immunostaining experiments. Each plasmid was co-transfected with minimal amount of pmaxGFP (Lonza) using Lipofectamine 2000 according to manufacturer recommendations.

**Cell fusion experiments.** Platinum E Cells (Cell Biolabs) were plated on a 100-mm culture dish at a density of $3 \times 10^6$ cells per dish and 24 h later were transfected with ten micrograms of retroviral plasmid DNA coding mutant *MYMK* and GFP protein using FuGENE 6 (Roche). Forty-eight hours after transfection, viral media was collected, passed through a 0.45-mm cellulose filter, and mixed with Polybrene (Sigma) at a final concentration of 6 μg ml$^{-1}$. For live staining experiments, 10T1/2 fibroblasts (~50–60% confluent on a 100-mm plate) were infected with 10 ml virus for 18 h. After infection, cells were washed × 3 with PBS, trypsinized, and mixed with C2C12 cells at a 1:1 ratio ($3 \times 10^5$ each cell type), and plated on a 35-mm dish in 10% FBS and DMEM. The following day the cells were placed in differentiation medium (DMEM with antibiotics and 2% Horse serum) for four DIV, and then GFP and myosin positive cells were analysed.

**Zebrafish husbandry.** All zebrafish experiments were approved by the Boston Children's Hospital Institutional Animal Care and Use Committee (IACUC). WT AB zebrafish strain was maintained according to the standard procedure of the Boston Children's Hospital Aquatic Resources Program. One male fish was crossed to two females to obtain enough fertilized eggs for each embryonic experiment. CRISPR-generated *mymk*$^{insT}$ fish were genotyped as described below and separated in different tanks according to their genotype. Homozygous mutant lines survive and are capable of self-feeding, but required separation from their heterozygous and WT siblings as young adults because they could not successfully compete for food and were at risk of starvation. The tanks housing *mymk*$^{insT}$ adult fish required cleaning once a month to avoid excessive accumulation of food at the bottom of the tank.

**Generation of *mymk*$^{insT}$ line.** Highly efficient CRISPR target sites for zebrafish *mymk* were selected using the browser CHOPCHOP (www.chopchop.cbu.uib.no)[50,51]. Two exon 3 (sgRNA1 and sgRNA2) targets and one exon 5 (sgRNA3) target were selected. sgRNAs were generated by annealing long oligos to

a Sp6 promoter sequence, followed by *in vitro* transcription[52]. To generate loss-of-function lines, 250 pg of sgRNA and 500 pg of recombinant Cas9 protein (PNA Bio Inc.) were incubated on ice for 10 min and injected into the cytoplasm of one-cell stage AB embryos as described for morpholino injections. Efficiency of the guide was evaluated by sequencing 10 of the approximately 100 injected embryos per clutch for each guide. Successful editing was indicted by multiple nucleotide peaks 4–5 bases 5′ to the PAM sequence. Chimeric sequences were identified for 60% of sgRNA1, 10% of sgRNA2 and 100% for sgRNA3 injected embryos. The low efficiency of editing for sgRNA2 may be explained by the presence of a SNP in our AB line located 5′ of the guide binding region. SgRNA1 guide was selected to establish the F0 population because sgRNA3 generated 100% edited embryos in the homozygous state, and the muscle hypoplasia of the resulting fish would place them at risk for decreased fertility secondary to impaired swimming and spawning.

F0 embryos generated with the sgRNA1 guide were outcrossed with WT AB and their offspring (F1) screened for germline transmission of the editing. We identified three different changes at the level of the cutting site, of which the most common was an insertion of a T at position c.434, predicted to cause a frame-shift resulting in 33 ectopic amino acids before a stop codon. Three founders were used to generate the F2 population, called *mymk^insT*, used in this study. Genotyping was performed using Taqman probes (Applied Biosystem) specifically designed to discriminate between the mutant *mymk^insT* and WT alleles. Primers used to generate the guides are as follows: sgRNA1 5′-GGGGTGTTGACCGCAGCTG TGA-3′; sgRNA2 5′-GGCATTTACTCCGGCCCCATCGG-3′; sgRNA3 5′-CGT GTCTCTGGCCATGTCCT-3′. Zebrafish exon 3 sequencing primers are as follows: Ex3FRW: 5′-TCAAAGTGGCTTAAAATGCTCA-3′; Ex3RVS: 5′-GAGTAGATG CCGTATCCCAGTC-3′.

**Western blot of overexpressed *MYMK* alleles in HeLa cells.** HeLa cells were collected at 24 h after transfection and lysed using RIPA buffer (ThermoFisher) supplemented with a cocktail of proteases and phosphatases inhibitors (Thermo-Fisher). After quantification of the total proteins using BCA assay, 10 μg of total lysate was loaded on a polyacrylamide gel 4–12% (Invitrogen) in Bis-Tris buffer for electrophoresis according to the manufacturer's recommendations. Novex nitro-cellulose membranes (Invitrogen) were used for transfer. The FLAG-tag signal was revealed using mouse monoclonal anti-FLAG M2 antibody (Sigma-Aldrich, F9291) at a final dilution 1:1,000, and anti-beta Actin HRP-conjugated (Abcam, ab20272) at final dilution of 1:10,000 was used as loading control.

**Histochemistry and immunofluorescence.** Human muscle biopsy sectioning (10 μm thick sections) and haematoxylin and eosin (H&E) staining for nuclei quantification were performed in the histology laboratory in the Department of Pathology at Boston Children's Hospital. Clinical diagnostic histochemical staining (12 μm thick sections) was performed in the Neuropathology laboratory in the Department of Pathology at Brigham and Women's Hospital: H&E, Myofibrillar ATPase at pH 4.3, Myofibrillar ATPase at pH 9.4, NADH diaphorase, Gomori trichrome, periodic acid-Schiff, and Oil Red O. Immunohistochemical staining (on 12 μm thick sections) was performed with the following primary antibodies: mouse: Pan-MHC (Developmental Studies Hybridoma Bank (DSHB), (A4.1025, 1:200), Slow-MHC (DSHB, A4.840, 1:100), Fast-MHC Type IIA (DSHB, SC-71, 1:100), MYH1A (DSHB, F59, 1:20) Embryonic MHC (DSHB, F1.652, supernatant, 1:18), Myogenin (DSHB, F5D, supernatant, 1:18), Pax7 (DSHB, Pax7, 1:50), Pecam-1/CD31 (DSHB, P2B1, 1:50), Lymphocyte Common Antigen (LCA) (DSHB, H5A5, 1:50), Vimentin (DSHB, 3CB2, supernatant, 1:44), Desmin (DSHB, D3, 1:100); rabbit: Laminin (Sigma-Aldrich, L9393, 1:1,000). The following secondary antibodies were used (1:1,000 dilution): goat anti-mouse highly-cross-adsorbed 546 (ThermoFisher, A-11030), goat anti-rabbit 488 (ThermoFisher, A-11008). Muscle sections were blocked and permeabilized for 2 h at room temperature (RT) with 10% normal goat serum in 0.3% Triton-X and 1X PBS, incubated overnight (ON) with primary antibodies in blocking solution at 4 °C, and stained with secondary antibodies for 2 h at RT, followed by counter-staining with DAPI (1:10,000) for 5 min, and mounting in Fluoromount-G. Imaging was performed on an Olympus VS120-SL whole slide scanner at 20× magnification.

For myosin staining of primary myoblasts, cells were plated on gelatin-coated permanox chamber slides (ThermoFisher), washed twice in PBS, and then fixed in 2% PFA in PBS. Cells were permeabilized with PBS + 0.5% Triton for 10 min at RT. After one hour incubation in blocking solution (2%BSA in PBS), cells were incubated overnight (ON) with primary antibody anti-myosin heavy chain all isoform (clone A4.1025, 1:100, DSHB) and then washed for three times with PBS followed by 1 h incubation with secondary fluorescent anti-mouse conjugated antibody. After three gentle 10 min washes in PBS, cells were incubated with DAPI for 5 min, washed, mounted with Fluoromount G, and imaged using an Olympus BX51 fluorescent microscope.

For live staining of HeLa cells (non-permeabilized), 24 h after transfection cells were washed once with ice cold PBS and blocked for 45 min in blocking buffer (3% BSA in PBS) on ice and after incubated for 1 h with anti-FLAG M2 in blocking buffer on ice. After 2 rinses with ice cold PBS, cells were fixed at RT with 4% PFA for 15 min, washed three times with PBS and incubated with conjugated secondary for 1 h at RT. Cells were then treated as described above. For staining of fixed cells, cells were treated as above with the only difference that cells were fixed at RT with

4% PFA for 15 min before the blocking and the primary incubation and not after. In this case all the incubations were performed at RT.

For phalloidin staining of zebrafish embryos, 24 hpf embryos were dechorionated with pronase and fixed with 4% sweet PFA (4% paraformaldehyde (PFA) in phosphate-saline buffer with 4% sucrose in PBS) for 3 h at RT. After two 5 min washes in PBS Tween 0.1% (PBSTween), embryos were permeabilized with PBS + Triton 2% and then stained ON with Alexa 488-conjugated phalloidin (ThermoFisher, A12379) diluted 1:100 in PBSTween. After four 10 min washes, embryos were incubated with DAPI in PBSTween for nuclear staining and additionally washed for 10 min. Embryos were manually deyolked and laterally mounted on slides with Fluoromount G (SouthernBiotech) for subsequent imaging. Embryos that were not oriented appropriately were excluded from analysis. Images were acquired using a confocal microscope LS710 with ZEN software (Carl Zeiss AG) and analysed using ImageJ software.

For myosin staining of zebrafish, dechorionated embryos were fixed in 4% sweet PFA and, after washing, were incubated in ice-cold methanol ON. After treatment with proteinase K for permeabilization, embryos were incubated one hour at RT with blocking solution (10% goat serum in PBS + 1.5% Triton X-100 (PBST)) and then ON with primary antibody against fast Myosin light chain (F310, DHBS 1:100) followed by six 30 min washes in PBST and ON incubation with 594 Alexa-conjugated goat anti-mouse antibody (1:500; ThermoFisher, A11032). After five 30 min washes in PBST, samples were treated, mounted, and images captured as described above.

For zebrafish histopathology, adult fish (five WT and three sex and age-matched mutants) were killed in ice-cold water, fixed in 4% PFA + 4% sucrose for 48 h, decalcified in 50 mM EDTA for 4 days, and then washed in PBS and serially dehydrated in ethanol. Paraffin embedding, sectioning (5 μm thick) and H&E staining of adult zebrafish were performed at Harvard Medical School Rodent Histopathology Core. Images were collected using a dual view Nikon Eclipse 90i at 2, 10 and 40× magnification, and analysed using ImageJ.

**RNA extraction and RT-PCR of *MYMK*.** Total RNA from 30 embryos was collected for MO4 injected and uninjected embryos using TRIzol (Invitrogen) according to manufacturer's specifications. RNA was then cleaned using RNeasy Mini kit (QIAGEN) and total RNA quantified by Nanodrop (ThermoFisher). 1.2 μg of total extracted RNA was retrotranscribed using SuperScript First-Strand Synthesis System (Invitrogen) with random hexamers to amplify any aberrant transcript that does not contain a poly(A) tail. The resultant cDNA was used for qualitative RT–PCR using the following three primers: z_mymk_E3, coding exon 3: 5′-CCGGAGTAGAATGCCGTATCC-3′; z_mymk_I2, intron 2: 5′-ACCTGTCAACATGCTCGTGAA-3′ for aberrant transcript; z_mymk_E5, coding exon 5: 5′-TTGACTAGCAGGGCATCGTG-3′.

Total RNA was extracted from mouse tissue or cultured cells with TRIZOL (Invitrogen) according to manufacturer instructions and contaminant DNA was removed. cDNA was synthesized using Superscript III reverse transcriptase with random hexamer primers (Invitrogen). Gene expression was analysed by qPCR using KAPA SYBR FAST (Kapa Biosystems). The following primers were used for *Mymk* analysis: F:5′-CCTGCTGTCTCTCCCAAG-3′, R:5′-AGAACCAGTGGGTC CCTAA-3′; 18S expression was used as a housekeeping gene. All analyses were performed on a 7900HT Fast Real-Time PCR machine (Life Technologies).

**Morpholino and mRNA injections.** Plasmids containing human WT and mutant *MYMK* coding sequence (described above) were linearized using NotI restriction enzyme and the linear product purified using phenol/chloroform. These sequences, containing the T7 promoter, were *in-vitro* transcribed using mMESSAGE mMA-CHINE T7 ULTRA (Ambion) according to manufacturer's conditions. This allowed for the incorporation of ARCA cap and poly(A) tail into the growing mRNA, generating a more stable RNA for injection in the embryo.

Four different morpholino sequences were designed and purchased from GeneTools. Morpholino oligonucleotide sequences MO1: 5′-ACTCACACAAA TGCCCTGGCGATTT-3′; MO2: 5′-ACTCAGAGAAATGGCCTCGGGATTT-3′; MO3: 5′-TCTTGGCGATAAACGCTCCCATTGC-3′; MO4: 5′-TTATTATTTG CTCACCCAGTAGCGT-3′. Three were translation-blocking morpholinos (MO1-3) of which two were previously published[13] and the fourth was a transcription-blocking morpholino targeting the donor site of exon 2 (Supplementary Fig. 6b). Stock solutions were diluted to 2 μg μl⁻¹ and optimal injection conditions were ascertained by a dose-response curve. An optimal final volume of 3 nl was injected into the embryo yolk at the one or two nuclei stage for each morpholino. Morpholino against p53 at a ratio of 3:2 was coinjected with each morpholino to reduce non-specific effects[53]. A control morpholino targeting the human beta-globin gene was used to test injection conditions. A minimum of 100 embryos were injected during each session; only a small increase (less than 10%) of embryo deaths was observed in morpholino injected versus uninjected embryos after 24 h. For rescue experiments, an optimal dose of 100 pg of mRNA generated from WT and mutant constructs (described above) were each independently co-injected with MO4 in or proximal to the nuclei of at least 100 embryos at the 1 cell stage. No evident toxicity was associated with mRNA injection.

**Rescue of *mymk$^{insT/insT}$* embryos.** For rescue of *mymk$^{insT/insT}$* zebrafish, 200 ng of WT or mutant mRNA was injected into the yolk of one-cell stage embryos obtained from F2 *mymk$^{insT/WT}$* crosses. To visualize the plasma membrane, 40 ng of membrane-bound mCherry was injected in the yolk of 8- to 16-cell stage of the same embryos to chimerically staining cell membranes. At least 100 embryos were injected with each allele for 3 or more replicate experiments, each on a separate day. We observed an average survival of 60% of the embryos after injection independent of the injected mRNA. Embryos were collectively fixed in PFA 4% in PBS and stained with DAPI. Each embryo was individually separated, the head was dissected for DNA extraction (using 50 mM NaOH stabilized with 10 mM Tris HCl pH 8) and genotyping, and the tail distal to the yolk sac was isolated in a single well of 96 well plate. After genotyping, tails of *mymk$^{insT/insT}$* embryos were mounted laterally in 80% glycerol and images acquired using a confocal microscope LS710 with ZEN software (Carl Zeiss AG). Analysis was blinded to the injected mRNA and scored according to the nuclear organization. Number of mutant embryos per genotype per injection were: *mymk$^{insT/insT}$* + mCherry = 3/4/4; *mymk$^{insT/insT}$* + WT = 7/7/8; *mymk$^{insT/insT}$* + M1 = 8/3/8/2; *mymk$^{insT/insT}$* + M5 = 2/5/5/2; *mymk$^{insT/insT}$* + M3 = 8/5/3/9; *mymk$^{insT/insT}$* + M2 = 2/3/7/10.

**Internuclear distance analysis.** To quantify the distance between nuclei, randomly selected images of 6 fish per allele (2 embryos from 3 different injections each on a different day) were opened in ImageJ and the *xy* position of each nucleus was annotated. The quantitative analysis of internuclear distances was performed using a custom written program on MATLAB. The program takes the coordinates of myonuclei as input, and gives the frequency distributions of the nucleus-nucleus distance *d*, for each phenotype. It then compares these distributions with each other and with the distribution produced by a random generated data set of *xy* positions. The 2D random distribution was obtained starting from 6 computer generated images, each with a number of objects equal to the average number and size of nuclei found in the zebrafish images (mean 110 nuclei, St.Dev. 20). To not overlap, any object with a distance $< 5\,\mu$m from any other was excluded, as zebrafish myonuclei are $\sim 5\,\mu$m. The histograms were normalized to $80\,\mu$m, as this is the peak, after which there is a decay due, in part, to the finite boundaries of the $200\,\mu$m image. Histograms of the same genotype were then averaged together and presented with SE bands. Deviations from the 2D random model were measured at distance *a* ($5–10\,\mu$m from the nucleus center that describes the nearest neighbor organization) and distance *b* ($18–28\,\mu$m from the nucleus center that describes the row organization), and statistical significance was tested between all distributions (6 distributions per genotype) using Two-Sample Kolmogorov-Smirnov test in MATLAB (kstest2). The code used for the analysis is available on request. Data were obtained from a minimum of two mutant embryos per clutch and three clutches from separate injections on different days.

**Statistical analyses and quantitation.** *Muscle biopsies.* Significance of fiber size (cross-sectional area and minimum Feret diameter) and number and nuclei number were assessed with Student's *t*-tests and ANOVA in Microsoft Excel. Graphs for the muscle biopsy analysis were generated in Microsoft Excel with mean ± standard deviation reported. $P < 0.05$ was considered significant.

*Fusion assay.* Analysis of fusion assay in transfected cells was performed on at least six fields per experiment for three replicates. For transfection efficiency the number of GFP-positive cells was counted for at least seven random fields per condition and in three independent experiments. The number of membrane positive FLAG cells was calculated in a similar fashion. For all cases significance was calculated using GraphPad imposing ordinary one-way ANOVA statistical analysis, using Bonferroni correction for multiple testing.

*Human myoblasts.* Analyses of differentiated myoblasts were performed on at least 10 random fields per experiment, and 3 independent experiments were performed. The fusion index was calculated as number of nuclei contained in myosin-positive cells. The differentiation index was calculated as the percentage of nuclei in myosin positive cells versus all nuclei in the field. Cell surface was calculated for 45 different cells for each condition and reported in micrometer. In both cases significance was calculated using two-tailed Student's unpaired *t*-test. For each data, mean ± s.e.m. is reported. $P < 0.05$ was considered significant.

*Zebrafish.* To determine the number of nuclei within each zebrafish embryo myotome, an area of $\sim 2,500\,\mu$m$^2$ close to sarcomere X was selected and outlined in ImageJ, and the number of cells (visualized by phalloidin staining) and nuclei inside each cell (visualized by DAPI staining) were counted. At least six independent embryos for each condition were randomly selected for non-blinded analysis from among those successfully mounted in the appropriate lateral orientation. All statistical analyses were performed using GraphPad Prism and differences between data were tested using two-tailed Student's unpaired *t*-test. $P < 0.05$ was considered significant. For each data, mean ± s.e.m. is reported.

For the qualitative analysis of nuclear organization in *tmem8c$^{insT/insT}$* rescued embryos, DAPI stained optical sections of injected embryos were blindly qualitatively separated in four categories based approximately to this scheme: no rescue = more than 90–100% of the nuclei in the figure were organized as the mCherry injected *mymk$^{insT/insT}$*; minimal rescue = 50–80% of the nuclei organized as in the mCherry injected *mymk$^{insT/insT}$* (some rescue, but overall more similar to mutant); partial rescue = 50–80% of the nuclei organized as in the mCherry

injected WT (not complete rescue but overall more similar to WT); rescued = 80–100% of nuclei organized as in the mCherry WT. Images were obtained by a minimum of two mutant embryos in at least three different days of injection.

**Data availability.** Data supporting the findings of this study are available from the authors on request. Genomic and phenotype data from a subset of participants are available through dbGAP with the Accession ID phs001383.v1.p1. Zebrafish line *mymk$^{insT/insT}$* (CL201) is registered at ZFIN with record number ZDB-ALT-170608-19.

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

## Acknowledgements

We thank the families for their participation; Jeff Stevens, Lisa Baird, Brith Otterud, Ying Hu, Hyun Cho, Christian Lawrence, the BCH Aquatic Resource Facility, the Harvard Medical School Rodent Histopathology core, and the HMS Ocular Genomics Institute for technical expertize and support; Matthew Alexander and Louis Kunkel for control myoblast cell lines; Sean Megason for kindly sharing the membrane-bound mCherry plasmid; Janbernd Kirschner for help with DNA sequencing; and Vandana Gupta, Umberto De Girolami, and Emanuela Gussoni for enlightening discussions. Research was supported by Swiss National Science Foundation P2LAP3_155081 and P300PA_164677 (SADG); 5T32GM007748 (MFR); Human Frontiers Science Program CDF LT000975/2015C (PA); The Slomo and Cindy Silvian Foundation, Inc. (BDW); P30 HD18655 (IDDRC, Boston Children's Hospital); Illumina, Inc., New Zealand Genomics Ltd., and Curekids (SPR); U01HD079068 (EWJ, IM, ECE, Moebius Syndrome Collaborative Research Group); NIH intramural programs of NHGRI, NEI, NIDCR, NINDS, NICHD, NIDCD, NIMH, and CRC; and Moebius Syndrome Foundation. ECE is an investigator of the Howard Hughes Medical Institute.

## Author contributions

E.C.E., S.R. and J.C.C. supervised, and S.A.D.G., S.C., N.M., W.M.C., M.L., M.B.S. and I.M.H. interpreted exome sequence. E.C.E, E.W.J., F.S.C. and C.G.B. supervised, and N.L.D.M.S., W.M.C., B.D.W., A.S. and P.S.C. performed targeted sequencing. I.M., E.C.E., J.C.C. and S.P.R. supervised, and C.V.R., C.R.F., C.D.R., T.H., C.E.S.-M., T.M. and D.M.M. and Moebius Syndrome Research Consortium members performed phenotyping studies. E.C.E. and C.G.B. supervised, and S.A.D.G., M.F.R., P.M. and C.G. performed human muscle biopsy and culture studies. S.A.D.G., E.C.E. and E.N.O. designed, and S.A.D.G., J.C. and A.R.M., performed cellular studies. S.A.D.G. and E.C.E. designed, and S.A.D.G., N.C. and L.C. performed zebrafish studies. S.A.D.G., J.C., M.R., P.A. M.F. and E.C.E. performed statistical analysis and quantitation. E.C.E. supervised the entire study. S.A.D.G., E.C.E. and I.M. wrote the manuscript, which was reviewed by all authors.

## Additional information

**Competing interests:** The authors declare no competing financial interests.

## Moebius Syndrome Research Consortium:

Caroline V. Andrews[1,2,3], Brenda J. Barry[1,2,3,12], David G. Hunter[26,27], Sarah E. Mackinnon[26], Sherin Shaaban[1,2,3], Monica Erazo[13,28], Tamiesha Frempong[29], Ke Hao[13,30], Thomas P. Naidich[31,32], Janet C. Rucker[33,34], Zhongyang Zhang[13,30], Barbara B. Biesecker[35], Lori L. Bonnycastle[16], Carmen C. Brewer[36], Brian P. Brooks[37], John A. Butman[38], Wade W. Chien[36], Kathleen Farrell[39], Edmond J. FitzGibbon[37], Andrea L. Gropman[40], Elizabeth B. Hutchinson[41,42,43], Minal S. Jain[39], Kelly A. King[36], Tanya J. Lehky[44], Janice Lee[45], Denise K. Liberton[45], Narisu Narisu[16], Scott M. Paul[39], Neda Sadeghi[41,42], Joseph Snow[46], Beth Solomon[39], Angela Summers[46], Camilo Toro[47], Audrey Thurm[46], Christopher K. Zalewski[36]

[28]Department of Obstetrics and Gynecology, Metropolitan Hospital, New York Health and Hospitals, New York, USA; [29]Department of Ophthalmology, Icahn School of Medicine at Mount Sinai, New York, USA; [30]Icahn Institute for Genomics and Multiscale Biology, Icahn School of Medicine at Mount Sinai, New York, USA; [31]Department of Radiology, Icahn School of Medicine at Mount Sinai, New York, USA; [32]Department of Neurosurgery, Icahn School of Medicine at Mount Sinai, New York, USA; [33]Department of Neurology, Icahn School of Medicine at Mount Sinai, New York, USA; [34]Department of Neurology, New York University School of Medicine, New York, USA; [35]Social and Behavioral Research Branch, National Human Genome Research Institute, National Institutes of Health, Bethesda, Maryland 20892-1477, USA; [36]Audiology Unit, Otolaryngology Branch, National Institute of Deafness and other Communications Disorders, National Institutes of Health, Bethesda, Maryland 20892-1477, USA; [37]Ophthalmic Genetics & Visual Function Branch, National Eye Institute, National Institutes of Health, Bethesda, Maryland 20892-1477, USA; [38]Radiology and Imaging Sciences, Clinical Center, National Institutes of Health, Bethesda, Maryland 20892-1477, USA; [39]Rehabilitation Medicine Department, Clinical Research Center, National Institutes of Health, Bethesda, Maryland 20892-1477, USA; [40]Division of Neurogenetics and Developmental Pediatrics, Children's National Medical Center, Washington, District of Columbia, USA; [41]Quantitative Medical Imaging Section, National Institute of Biomedical Imaging and Bioengineering, National Institutes of Health, Bethesda, MD, USA; [42]Section on Quantitative Imaging and Tissue Sciences, Eunice Kennedy Shriver National Institute of Child Health and Human Development, National Institutes of Health, Bethesda, Maryland, USA; [43]The Henry M. Jackson Foundation for the Advancement of Military Medicine, Inc, Bethesda, Maryland, USA; [44]EMG Section, National Institutes of Neurological Disorders and Stroke (NINDS), National Institutes of Health, Bethesda, Maryland 20892-1477, USA; [45]Craniofacial Anomalies and Regeneration Section, National Institute of Dental and Craniofacial Research, National Institutes of Health, Bethesda, Maryland 20892-1477, USA; [46]Office of the Clinical Director, National Institute of Mental Health, National Institutes of Health, Bethesda, Maryland 20892-1477, USA; [47]NIH Undiagnosed Diseases Network (UDN), Common Fund, National Human Genome Research Institute, National Institutes of Health, Bethesda, Maryland 20892-1477, USA.

