## [Peer Review File · Nature Communications]

Reviewers' comments:

Reviewer #1 (Remarks to the Author):

Review of the manuscript, "A defect in myoblast fusion underlies Carey-Fineman-Ziter syndrome".
Authored by Di Gioia et al.

The team found autosomal recessive mutations in TMEM8C cause CFZS by a loss of function mechanism. This was discovered by rare variant analysis of WES performed for all members of a set of nuclear families each with 2 affected siblings. All were unrelated, some from different countries (US, New Zealand, Brazil). All affected individuals had mutations in this gene in common. Some of the mutations occurred in more than one family. The disease phenotype was well described, which included histology sections from muscle biopsies. It was previously known that mice with heterozygous null Tmem8c alleles are normal, while homozygous null mutant mice die perinatally. The variants were validated for loss of function or as hypomorphic alleles in cell culture models and zebrafish.

This is a very well written manuscript that describes the cause CFZS and follows up with extensive phenotypic and biological validation studies. There are some minor questions that can be addressed in the writing.

- 1) Could the authors comment in the paper on the possible genetic heterogeneity of CFZS/related syndromes since 1% appears to harbor mutations in this gene based upon analysis of 300 additional probands. Were other variants found in TMEM8C but were not predicted as damaging or not as rare as these? Did the affected individuals in Families 4+5 look phenotypically more similar to Families 1-3 than the other 300 probands?
- 2) Some of the defects in the cells in Figures 3b and 4d were difficult to see; perhaps some better labeling of the images would help.
- 3) The only zebrafish experiment that would have been better than what they did was to use CRISPR to knockin the individual mutations.

Reviewer #2 (Remarks to the Author):

The manuscript by Di Gioia et al. entitled “A defect in myoblast fusion underlies Carey-Fineman-Ziter syndrome” describes the identification of 8 individuals (from 5 families) with bi-allelic changes in TMEM8C. By functional analysis, the authors showed that mutations M1 and M5 are hypomorphic while M2, M3 and M4 are null, and that the normal function of TMEM8C is necessary for muscle development and maintenance in humans. Both genetic and functional data are convincing.

Minor points:

1. Since individual 8 carries the homozygous M5 hypomorphic variant, is the patient expected to have a less severe phenotype?
2. The variant nomenclature for M4, “p.0” should be “p.0?” since no experimental data are available to show that the elimination of the start codon results in no protein production.
3. Line 196 on page 11: reference number for Millay et al. was listed as 9, which appears to be incorrect (Reference 9 on page 22 is Kim et al.).
4. Supplementary table 2 (page 16): M4 and M4 appear to be switched.

Reviewer #3 (Remarks to the Author):

Engle and colleagues report on 8 affected individuals from 5 families with Carey Fineman Ziter syndrome. Using WES, they have identified recessive variants in TMEM8C associated with disease in these families. They use in vitro studies complemented by zebrafish work to validate these variants.

Overall, the data is convincing that TMEM8C is the cause of CFZS in these families. The strongest data is from the human genetics, where the variants are rare, likely pathogenic, and segregate with disease. There is also no other clear candidate from the WES data. Lastly, the potential function of TMEM8C fits with many aspects of the overall clinical phenotype. The supporting validation data is less convincing. The C2C12 fusion experiments are the most compelling, and show that the deleterious variants are not able to promote myoblast fusion to the extent the WT TMEM8C can. The patient myocyte study is greatly limited by the fact that it is only a single individual. Lastly, the fish study is only somewhat convincing, particularly because (a) knockdown does not appear to be efficient with their morpholinos and (b) the myofibers are not adequately outlined for analysis.

Specific major comments:

(1) There is well documented variability in fusion and myotube formation in patient derived myocytes that is mutation independent. Therefore, while it is suggestive that impaired fusion is seen in the patient cells, it is not adequate to make significant conclusions from one patient, despite how many cellular replicates are observed from that one patient. While it is understood that only one individual has had a muscle biopsy, only sources for generating patient myotubes could be explored (transdifferentiation from patient fibroblasts for example).

(2) The zebrafish studies are difficult to interpret, though in general support the overall conclusions regarding the deleterious nature of the TMEM8C variants. The morpholinos are not associated by western with complete knockdown, and there appears to be quite a bit of residual protein remaining. Also, the splice site morpholino appears to achieve very modest knockdown, with abundant normal length transcript remaining. Because of the significant residual protein in all cases, it would be important to know how much variability is seen in knockdown with independent clutches and injections. Also, it would be important to demonstrate that protein levels are reduced for each morpholino in a dose dependent fashion.

In terms of the nuclear counts in the fish, there are several considerations. One is that phalloidin is not a standard marker of the sarcolemmal membrane, and rather more typically a marker of the sarcomere itself. Therefore it seems possible that nuclear number per fiber is not adequately captured using this as the marker. In addition, the count numbers are very small for morpholino based experiments, with max 8 animals counted per condition. It would be helpful to understand whether these were measured from different clutches and injections, as it is critical to observe such data points from at least 3 independent clutches.

Minor comments:

(1) the muscle biopsy does not look like classic CFTD. The small type I fibers appear angulated and look more neurogenic than anything. The type II hypertrophy is much greater than is typically seen. Combined with the facts that (a) the clinical phenotype contains some non-myopathic features and (b) the gene is expressed (at least in GTex) in peripheral nerve, how convinced are the authors that there is not an element of neuropathy or distal SMA-like features in this disease?

(2) the pictures shown in the main figure make the facial weakness very hard to appreciate. Is there a reason why the compelling images in the supplemental figure are not presented in the main figure?

(3) what are the authors explanation for the micrognathia and distinctive dysmorphisms that would be considered atypical for even severe congenital myopathies.

(4) did the authors consider blocking UPS as a means for proving that the reason they see less protein in the C2C12 cells is due to degradation?

(5) phalloidin is not an antibody per se.

(6) did any of the patients have brain MRI? There are reports of abnormal brain MRI in some patients with CFZS, which would not necessarily fit with the known expression and function of TMEM8C.

REVIEWERS' COMMENTS:

Reviewer #3 (Remarks to the Author):

This is a revised manuscript by Engle and colleagues describing TMEM mutations as a cause of CFZS. The authors did a really excellent job responding to the reviewer critiques, and have strengthened what was already a very interesting and informative manuscript. In particular, the new Table of clinical features really underscores the shared phenotype in this genetic subtype of CFZS, and the zebrafish studies are very compelling and elegant.

Reviewer #1:

The team found autosomal recessive mutations in TMEM8C cause CFZS by a loss of function mechanism. This was discovered by rare variant analysis of WES performed for all members of a set of nuclear families each with 2 affected siblings. All were unrelated, some from different countries (US, New Zealand, Brazil). All affected individuals had mutations in this gene in common. Some of the mutations occurred in more than one family. The disease phenotype was well described, which included histology sections from muscle biopsies. It was previously known that mice with heterozygous null Tmem8c alleles are normal, while homozygous null mutant mice die perinatally. The variants were validated for loss of function or as hypomorphic alleles in cell culture models and zebrafish.

This is a very well written manuscript that describes the cause CFZS and follows up with extensive phenotypic and biological validation studies. There are some minor questions that can be addressed in the writing.

1) Could the authors comment in the paper on the possible genetic heterogeneity of CFZS/related syndromes since 1% appears to harbor mutations in this gene based upon analysis of 300 additional probands. Were other variants found in TMEM8C but were not predicted as damaging or not as rare as these? Did the affected individuals in Families 4+5 look phenotypically more similar to Families 1-3 than the other 300 probands?

We have addressed these questions below (modifying the order) and added text to the paper as follows:

C. Did the affected individuals in Families 4+5 look phenotypically more similar to Families 1-3 than the other 300 probands?

To address this question and to clarify the CFZS phenotype, we have modified Table 1 to now highlight (a) the 7 clinical features shared in common by all 8 mutation-positive individuals followed by 12 other features in declining incidence (last column), and the number of these features that each individual has (new row entitled "Clinical score") and (b) the 3 laboratory examinations with common findings in all participants tested (EMG, muscle MRI and brain MRI, CPK), and a new bottom row entitled "Total score". This revised table permits the reader to appreciate more easily that individuals 7 and 8 have a similar total score to the other affected individuals. While individual 8 is the only participant who did not have feeding problems and growth failure necessitating a gastro/jejunostomy, two others did not require gastro/jejunostomy, and we are not convinced that this distinguishes her significantly from the others.

A. Could the authors comment in the paper on the possible genetic heterogeneity of CFZS/related syndromes since 1% appears to harbor mutations in this gene based upon analysis of 300 additional probands.

To begin to define the phenotypic variability of CFZS, *TMEM8C* was sequenced in >300 probands who had some phenotypic overlap with the mutation-positive individuals identified by exome sequencing and with the cases diagnosed as CFZS described in the literature (which are now also mentioned in the introduction). We felt screening a broader spectrum of phenotypes was indicated given these reports, as well as the recent report of *PLXND1* and *REV3L* variants in probands with variable phenotypes ranging from isolated facial palsy and classic Moebius to complex syndromes including facial palsy, Poland, limb and cardiac

defects (Tomas-Roca et al, Nature Communications, 2015).

We now state in the text that the >300 probands we screened included the following pre-molecular diagnoses: 1 with CFZS, ~10 with a congenital myopathy with predominant involvement of facial musculature, ~50 with isolated or syndromic congenital facial weakness (and normal eye movements), and ~250 with Moebius syndrome. While Moebius syndrome diagnostic criteria include facial weakness, complete or virtually complete limitation of eye abduction, and intact vertical gaze with or without additional clinical findings, many clinicians use this diagnosis as a catch-all for individuals with isolated or syndromic facial weakness; for that matter, several of the *TMEM8C*-CFZS mutation positive individuals had been diagnosed with Moebius syndrome prior to their enrollment in research. The two individuals from this cohort of >300 who were found to harbor mutations by Sanger sequencing included the individual diagnosed clinically with CFZS and one individual diagnosed with Moebius syndrome; on examination at NIH, the latter was found to have normal eye movements. Both fit the diagnostic criteria of CFZS (see description above). While we are not able to fully score the >300 mutation negative probands according to table 1, they appear to be **clinically distinguishable** from the *TMEM8C*-mutation positive probands. We agree, however, that while the CFZS phenotype appears distinct and discrete, additional deep phenotyping and sequencing will be necessary to define the full spectrum of *TMEM8C* mutations and the potential heterogeneity of CFZS.

We have rewritten the text to provide greater clarity, and have added details about the potential slow progression of weakness and respiratory insufficiency that we believe is associated with CFZS in the results and discussion.

B. Were other variants found in TMEM8C but were not predicted as damaging or not as rare as these?

We identified four synonymous coding variants and five intronic variants. All are known polymorphisms and none were found in combination with other compound heterozygous changes. These SNPs are now included in a table in the Supplement.

2) Some of the defects in the cells in Figures 3b and 4d were difficult to see; perhaps some better labeling of the images would help.

We have altered the contrast, included arrows/arrowheads, and altered the order of the photomicrographs to enhance the visualization of cell-cell fusion.

3) The only zebrafish experiment that would have been better than what they did was to use CRISPR to knockin the individual mutations.

We are pleased to now include a *tmem8c* zebrafish mutant line that we developed using CRISPR/Cas technology; please refer to our more detailed response regarding these functional studies provided below with regard to Reviewer 3, point 2. These data are presented in new Figures 5, 6, and S7, and the morpholino data has been moved to the Supplement as Figure S6.

Reviewer #2:

The manuscript by Di Gioia et al. entitled “A defect in myoblast fusion underlies Carey-Fineman-Ziter syndrome” describes the identification of 8 individuals (from 5 families) with bi-allelic changes in TMEM8C. By functional analysis, the authors showed that mutations M1 and M5 are hypomorphic while M2, M3 and M4 are null, and that the normal function of TMEM8C is necessary for muscle development and maintenance in humans. Both genetic and functional data are convincing.

Minor points:

1. Since individual 8 carries the homozygous M5 hypomorphic variant, is the patient expected to have a less severe phenotype?

Please refer to our response to Reviewer 1 point 1. Individual 8 has a slightly attenuated phenotype, in that she did not present with the growth failure and feeding difficulties necessitating a gastrostomy tube, but has a similar and slightly higher clinical score (15/20 features) than individuals 3 and 4. Thus, we do not feel that her phenotype is significantly different than the other affected individuals. This supports our statement in the discussion that “We demonstrate that recessive mutations in *TMEM8C* cause CFZS through a combination of one hypomorphic and one null allele, or two hypomorphic alleles, which reduce *TMEM8C* function and myoblast fusion below a threshold but not to zero.”

Our data do support greater loss of *tmem8c* function with the M5 allele (which individual 8 has in the homozygous state) than with the M1 allele (which other have in combination with a null allele). Thus, the final protein level may be similar, and/or similar phenotypes may result when the protein level is reduced to within a broader range. In the future we hope to determine whether M1 in the homozygous state (which is present in one individual in the EXAC database) causes a mild phenotype, and establish whether two null mutations in humans are compatible with life (which we suspect are not).

2. The variant nomenclature for M4, “p.0” should be “p.0?” since no experimental data are available to show that the elimination of the start codon results in no protein production.

Thank you for picking up this error and we have made the change in the text and figures.

3. Line 196 on page 11: reference number for Millay et al. was listed as 9, which appears to be incorrect (Reference 9 on page 22 is Kim et al.).

Thank you, we have corrected this error and double-checked the references for accuracy.

4. Supplementary table 2 (page 16): M4 and M4 appear to be switched.

Thank you for picking up this error and we have corrected this in the revised supplemental material.

Reviewer #3:

Engle and colleagues report on 8 affected individuals from 5 families with Carey Fineman Ziter syndrome. Using WES, they have identified recessive variants in TMEM8C associated with disease in these families. They use *in vitro* studies complemented by zebrafish work to validate these variants.

Overall, the data is convincing that TMEM8C is the cause of CFZS in these families. The strongest data is from the human genetics, where the variants are rare, likely pathogenic, and segregate with disease. There is also no other clear candidate from the WES data. Lastly, the potential function of TMEM8C fits with many aspects of the overall clinical phenotype. The supporting validation data is less convincing. The C2C12 fusion experiments are the most compelling, and show that the deleterious variants are not able to promote myoblast fusion to the extent the WT TMEM8C can. The patient myocyte study is greatly limited by the fact that it is only a single individual. Lastly, the fish study is only somewhat convincing, particularly because (a) knockdown does not appear to be efficient with their morpholinos and (b) the myofibers are not adequately outlined for analysis.

Specific major comments:

(1) There is well documented variability in fusion and myotube formation in patient derived myocytes that is mutation independent. Therefore, while it is suggestive that impaired fusion is seen in the patient cells, it is not adequate to make significant conclusions from one patient, despite how many cellular replicates are observed from that one patient. While it is understood that only one individual has had a muscle biopsy, only sources for generating patient myotubes could be explored (transdifferentiation from patient fibroblasts for example).

We appreciate the reviewer's concerns and agree with them. Unfortunately, we have been unable to obtain myocytes from other probands, and do not have patient fibroblasts for transdifferentiation. We consider the data of interest and consistent with our functional data and that of others in mouse and zebrafish, and so we have softened our statement in the text and left the data in as a portion of figure 5. If you and the editor prefer, they can be moved to the supplement.

(2) The zebrafish studies are difficult to interpret, though in general support the overall conclusions regarding the deleterious nature of the TMEM8C variants. The morpholinos are not associated by western with complete knockdown, and there appears to be quite a bit of residual protein remaining. Also, the splice site morpholino appears to achieve very modest knockdown, with abundant normal length transcript remaining. Because of the significant residual protein in all cases, it would be important to know how much variability is seen in knockdown with independent clutches and injections. Also, it would be important to demonstrate that protein levels are reduced for each morpholino in a dose dependent fashion.

In terms of the nuclear counts in the fish, there are several considerations. One is that phalloidin is not a standard marker of the sarcolemmal membrane, and rather more typically a marker of the sarcomere itself. Therefore it seems possible that nuclear number per fiber is not adequately captured using this as the marker. In addition, the count numbers are very small for morpholino based experiments, with max 8 animals counted per condition. It would be helpful to understand whether these were measured from different clutches and injections, as it is critical to observe such data points from at least 3 independent clutches.

We have addressed the potentially confounding effect of the residual *tmem8c* protein following morpholino knockdown by providing new data from our *tmem8c*^{insT/insT} zebrafish loss-of-function model that we generated by CRISPR. We find that F2 *tmem8c*^{insT/insT}

embryos have no fusion of type 2 muscle fibers, and at 48 hpf their myofiber nuclei align in the middle of the sarcomere, resulting in a very clear and 'scorable' phenotype.

Tmem8c^{insT/insT} zebrafish can survive to adulthood (likely because of the preservation of type 1 slow fibers that do not fuse in wildtype fish) but by 3 months of age are smaller than wildtype fish, have an abnormal angle to their jaw, and have smaller-appearing type 2 muscle fibers. Introduction of *tmem8c*WT-mRNA into *tmem8c*^{insT/insT} embryos successfully rescues myoblast fusion in ~40% of fish, and partially rescues in another 40%. By contrast, attempted rescue with the null mutations fails (0%) and with the hypomorphic mutations is intermediate (~10% rescue, 30% partial rescue, 40-50% minimal rescue, and 10-20% no rescue). These data are evaluated both qualitatively and quantitatively, and the differences are statically significant. For each mutation studied, embryos were examined from at least 3 independent clutches and injected on three different days. We now present these new data in Figures 5, 6 and S7, and have moved the morpholino data to the supplement as supporting evidence, along with numbers of clutches, injections, and embryos analyzed (Supplemental Figure S6).

Of note, we have now tested four different anti-TMEM8C antibodies and, despite much troubleshooting, none work in zebrafish by Western blot or immunocytochemistry. We are, however, confident that our model is loss-of-function based on (1) the similar but more severe phenotype than that which we find after morpholino knockdown, (2) our ability to rescue the phenotype with wildtype overexpression, and finally, (3) two other alleles that truncate *tmem8c* at amino acid residues 31 and 59 that were published this month and have the identical phenotype and the fish also survive (W Zhang and S Roy, Dev Biol, 2017). Moreover, because we are no longer confident of our original antibody we used to assess the morpholino knock-downs, we have removed that western blot from the supplement.

In these new studies, we inject mCherry mRNA for visualization at 48 hpf. Our new method of statistical analysis does not require the counting of nuclei, but instead the method assesses their relative positions.

Minor comments:

(1) the muscle biopsy does not look like classic CFTD. The small type I fibers appear angulated and look more neurogenic than anything. The type II hypertrophy is much greater than is typically seen. Combined with the facts that (a) the clinical phenotype contains some non-myopathic features and (b) the gene is expressed (at least in GTex) in peripheral nerve, how convinced are the authors that there is not an element of neuropathy or distal SMA-like features in this disease?

We appreciate the reviewer's concern, and have edited the text, table, and figures to better reflect the data we have in support of a myogenic etiology for CFZS. We state that the probands have no fasciculations or clinical signs of peripheral nerve involvement and EMG/NCS studies obtained in 4 affected participants are myopathic without neurogenic changes. We provide a summary of the EMG of individual 2 in the legend to Table 1. Consistent with a myopathic phenotype, we include mRNA expression in peripheral nerve and muscle, developing muscle, and injured muscle in Figure S3. While GTex does report expression in adult peripheral nerve, we did not detect expression in the sciatic nerve.

The biopsy shows a disproportionate number of type I fibers and very enlarged type II fibers. The great size differential between the type I and type II fibers in this biopsy are remarkable and not typically seen in neurogenic disorders, whereas the small type I fibers are a characteristic feature of congenital myopathies. We agree that the muscle biopsy does show

some clustering of variably-sized type I fibers, many of which are small and polygonal/angulated. Denervation and reinnervation typically affect both fiber types, however, and type II fibers do not show reciprocal clustering. We also did not identify target fibers, which are commonly seen in neurogenic disorders, or the characteristic SMA pattern of small rounded fibers in large groups. Hence, although there is clustering of small type I fibers, the overall assessment best fits a myogenic process.

These findings currently fit best within the definition of congenital fiber type disproportion myopathy (CFTD): small type one fibers combined with a disproportionate number of type I vs type II fibers. We have expanded the discussion of this concept, placed these findings in the context of similar clinical syndromes, and state that future studies may lead to the recategorization of CFZS.

(2) the pictures shown in the main figure make the facial weakness very hard to appreciate. Is there a reason why the compelling images in the supplemental figure are not presented in the main figure?

We now present full face and profile photos of the faces of each individual in the main figure to demonstrate the facial weakness more convincingly.

(3) what are the authors explanation for the micrognathia and distinctive dysmorphisms that would be considered atypical for even severe congenital myopathies.

Facial dysmorphism, including Robin sequence, high palate, dolichocephaly, as well as fetal akinesia deformation sequence/arthrogryposis phenotypes are common in a number of congenital myopathies. They are particularly common in *MTM1*-related centronuclear myopathy, severe congenital nemaline myopathy, severe neonatal *DNM2*-related and severe *RYR1*-related myopathies, particularly those associated with recessive inheritance (for review see North KN et al 2014). Cleft palate is uncommon and is only observed in the two original siblings –Individuals 1 and 2 described by Dr. Carey, but has been described in patients with Native American myopathy (*STAC3*, AR), Marden/Walker and Gordon syndrome or distal arthrogryposis type 3 (*PIEZO2*, AD), and early-onset myopathy, areflexia, respiratory distress, and dysphagia (EMARDD) caused by homozygous or compound heterozygous mutation in the *MEGF10* gene, among others. Dr. Carey provided a detailed discussion of the mechanisms of Robin sequence in the context of a congenital myopathy in his original and follow-up manuscripts on CFZS (Carey et al 1982 and 2004).

We have now added to the discussion a brief summary of the CFZS in the larger context of congenital myopathies, and we have added a paragraph about both zebrafish and human selective vulnerability in this disorder.

(4) did the authors consider blocking UPS as a means for proving that the reason they see less protein in the C2C12 cells is due to degradation?

We appreciate this suggestion. We had tried to assess degradation prior to the first submission and again during the revision period, and unfortunately the results have been inconclusive. We have tried transfecting HeLa cells with wildtype and mutant constructs and inhibiting the proteasome with MG132 or lysosomes with chloroquine, and we do not observe more of an increase in mutant protein in the soluble fraction comparable to the increase in WT. We have also overexpressed the constructs in HEK293 cells. Following a long exposure

time, we see a minimal amount of M1- or M5- (hypomorphic) protein and even less M2-M4 (LOF) protein in the soluble fraction, consistent with protein aggregation (see Figure S5). We have tried extracting the insoluble fraction by sonication but again fail to see the mutant product. We feel these data reflect, in part, our failure to fully extract the mutant protein from the insoluble fraction, and in part lower expression of the mutant constructs compared to WT; this may reflect poor folding of the mutant proteins and/or use of additional degradation pathways that we did not explore. After considerable effort and inconclusive results, we have chosen to soften our statement in the text and exclude these experiments, as we agree they are a minor point.

(5) phalloidin is not an antibody per se.

We thank the reviewer for picking up this error, and have corrected it.

(6) did any of the patients have brain MRI? There are reports of abnormal brain MRI in some patients with CFZS, which would not necessarily fit with the known expression and function of TMEM8C.

Individuals 2, 5 and 7 underwent brain MRI, and we detected no abnormalities in the brainstem or brain parenchyma. We now include representative MRI images in Fig. S2. These findings are in contrast to cases reported to have clinical CFZS who had ventriculomegaly, white matter changes in the frontal lobes and basal ganglia, and brainstem and pontine hypoplasia or heterotopias (Ryan et al 1999, Verloes et al 2004, Dufke et al 2004, and Pasetti et al 2016). Dr. Carey has previously challenged the clinical diagnosis of these individuals as CFZS (Carey et al 2004) and we have reached out to authors to molecularly confirm their diagnoses (in process). Based on our current understanding of the clinical and molecular findings, individuals with cranial nerve palsies (CCDDs), intellectual disabilities and/or brain MRI anomalies do not have CFZS resulting from *TMEM8C* mutations. Testing more individuals for *TMEM8C* mutations will help to continue to delineate the phenotypic spectrum of this new congenital myopathy.

We now introduce and discuss these reports, highlighting how they differ from *TMEM8C*-CFZS.

Response to reviewer

Reviewer #3 (Remarks to the Author):

This is a revised manuscript by Engle and colleagues describing TMEM mutations as a cause of CFZS. The authors did a really excellent job responding to the reviewer critiques, and have strengthened what was already a very interesting and informative manuscript. In particular, the new Table of clinical features really underscores the shared phenotype in this genetic subtype of CFZS, and the zebrafish studies are very compelling and elegant.

We sincerely appreciate the reviewer positive and complementary comments.